# ArcVQ-VAE: A Spherical Vector Quantization Framework with ArcCosine Additive Margin

**Jaeyung Kim** [1 2]  **YoungJoon Yoo** [1 2]

## Abstract

Vector Quantized Variational Autoencoder (VQ-VAE) has become a fundamental framework for learning discrete representations in image modeling. However, VQ-VAE models must tokenize entire images using a finite set of codebook vectors, and this capacity limitation restricts their ability to capture rich and diverse representations. In this paper, we propose ArcCosine Additive Margin VQ-VAE (ArcVQ-VAE), a novel vector quantization framework that introduces a spherical angular-margin prior (SAMP) for the codebook of a conventional VQ-VAE. The proposed SAMP consists of Ball-Bounded Norm Regularization, which constrains all codebook vectors within a time-dependent Euclidean ball, and ArcCosine Additive Margin Loss, which encourages greater angular separability among latent vectors. This formulation promotes more discriminative and uniformly dispersed latent representations within the constrained space, thereby improving effective latent-space coverage and leading to improved codebook utilization. Experimental results on standard image reconstruction and generation tasks show that ArcVQ-VAE achieves competitive performance against baseline models in terms of reconstruction accuracy, representation diversity, and sample quality. The code is available at: https://github.com/goals4292/ArcVQ-VAE

## 1. Introduction

Image tokenization based on Vector Quantization (VQ) has become a key component in a wide range of visual

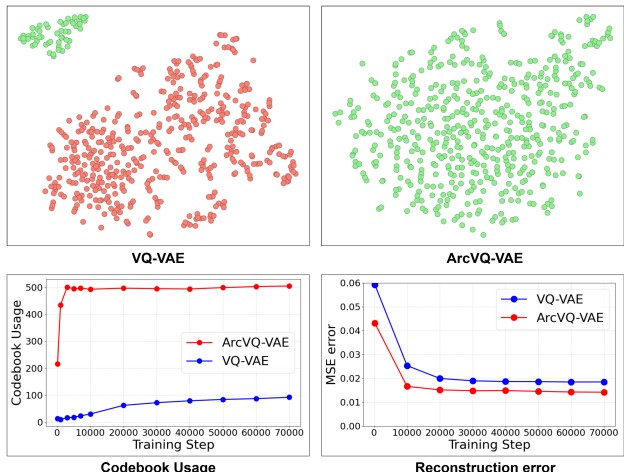

*Figure 1.* **t-SNE visualizations of the codebook vector distributions** (top) and **quantitative comparisons of codebook usage and reconstruction error** (bottom) between VQ-VAE and ArcVQ-VAE. In the t-SNE plots, green points indicate codebook vectors that are activated during inference, while red points represent inactive vectors. ArcVQ-VAE exhibits more uniformly dispersed codebook entries in the latent space, higher codebook usage, and lower reconstruction error.

tasks, including image compression (Agustsson et al., 2017; Van Den Oord et al., 2017; Williams et al., 2020), generation (Razavi et al., 2019; Esser et al., 2021; Yu et al., 2021; Lee et al., 2022; Zheng et al., 2022), and understanding (Liu et al., 2022; Mao et al., 2021; Zhang et al., 2024; Guotao et al., 2024; Sargent et al., 2023; Ma et al., 2025). Specifically, the Vector Quantized Variational Autoencoder (VQ-VAE) (Van Den Oord et al., 2017) discretizes continuous embedding space into a finite set of codebook vectors through a quantization process, enabling effective learning of compact and expressive image representations.

However, VQ-VAE models face several challenges. Since a finite set of codebook vectors must tokenize an entire image dataset, the limited capacity of the codebook constrains its ability to capture rich and diverse representations. Furthermore, codebook collapse, a phenomenon in which only a small subset of codebook vectors is actively utilized during quantization, while the majority remain unused, amplifies the problem (Kaiser et al., 2018; Takida et al., 2022).

---

[1]Department of Artificial Intelligence, Chung-Ang University, Seoul, Republic of Korea [2]SNUAILAB, Seoul, Republic of Korea. Correspondence to: Jaeyung Kim <goals4292@cau.ac.kr>, Youngjoon Yoo <yjyoo3312@cau.ac.kr>.

*Proceedings of the $43^{rd}$ International Conference on Machine Learning*, Seoul, South Korea. PMLR 306, 2026. Copyright 2026 by the author(s).

To enable each codebook vector to capture richer information and to address the codebook collapse issue, various approaches (Zheng & Vedaldi, 2023; Zhu et al., 2024; Takida et al., 2022; Vuong et al., 2023) have been proposed, including online K-means clustering (Zheng & Vedaldi, 2023), pretrained embeddings (Zhu et al., 2024), stochastic quantization with posterior modification (Takida et al., 2022), and the application of the Wasserstein distance to the discrete codebook (Vuong et al., 2023). Nevertheless, most of these methods focus on the frequency or mechanism of codebook selection and update, either with deterministic and stochastic approaches, without sufficiently addressing the geometric imbalance of the latent space.

In this paper, we propose **ArcVQ-VAE**, a novel vector quantization framework that introduces a spherical angular-margin prior (SAMP) on the codebook, combining Ball-Bounded Norm Regularization and ArcCosine Additive Margin Loss to resolve this structural limitation. The proposed scheme acts as a prior constraint on the codebook vectors, requiring each one to reside within a time-dependent Euclidean ball, while encouraging latent vectors to be angularly separated. This promotes more discriminative and uniformly dispersed representations, improving the effective coverage and utilization of the codebook. The extensive experiments on public benchmarks demonstrate that ArcVQ-VAE achieves a much higher codebook utilization rate than conventional VQ-VAE and delivers superior reconstruction and generation performance, even capturing delicate local details. Finally, our contributions are summarized as follows:

- We propose ArcVQ-VAE, a novel vector quantization framework that incorporates the geometry of the latent space by enforcing a spherical angular-margin prior (SAMP) between the codebooks in a unified probabilistic framework. This design encourages more discriminative and uniformly dispersed latent representations within a Euclidean ball, which improves codebook usage and representational quality.

- The proposed ArcVQ-VAE method incurs virtually no additional computational cost and does not introduce any extra network components. Simply applying norm-based scaling to the codebook vectors after each training batch and incorporating a lightweight angular margin loss term is all we need for implementing SAMP of ArcVQ-VAE.

- Through comprehensive experiments, we demonstrate that the proposed ArcVQ-VAE consistently outperforms baseline methods in terms of reconstruction accuracy, representation diversity, and overall sample quality, validating the effectiveness of our approach.

## 2. Related Works

**Vector Quantized Codebook.** After the proposal to discretize the latent embedding of a variational autoencoder (Kingma et al., 2013) into a finite set of codes (Van Den Oord et al., 2017), the discretization technique has become a fundamental building block for modern generative models such as autoregressive (Razavi et al., 2019; Esser et al., 2021) or diffusion (Rombach et al., 2022). Building on this discretization-based representation learning, subsequent work on codebook learning (Razavi et al., 2019; Zheng & Vedaldi, 2023; Zhu et al., 2024; Takida et al., 2022; Vuong et al., 2023; Wang et al., 2025; Yoo & Choi, 2024) has emerged as an active research direction, aiming to endow the codebook with richer semantic content while promoting the utilization of the available codebook entries. Razavi et al. (2019) introduces a multi-scale, hierarchical codebook that simultaneously handles both fine details and global context of an image. Vuong et al. (2023) proposes a Wasserstein distance metric (Tolstikhin et al., 2017) to address distance measurement in discrete vector spaces. Takida et al. (2022) modifies the posterior distribution of the VQ-VAE to be more robust to the codebook collapse problem. Zheng & Vedaldi (2023) applies an adaptive K-means clustering (Arthur & Vassilvitskii, 2006) method that relocates underutilized codebook vectors closer to the encoded feature vectors, thereby encouraging their utilization. Zhu et al. (2024) leverages a pretrained vision encoder to extract codebook vectors for effective codebook scale-up.

**Angular Margin Loss and Hyperspherical Learning.** Imposing desired prior information on vectors in high-dimensional spaces is a classical challenge in the field of metric learning. Since the introduction of deep contrastive learning (Chopra et al., 2005) and triplet loss (Schroff et al., 2015), a large body of subsequent research has emerged. Among these, approaches that normalize all cluster or class vectors onto a unit hypersphere and maximize the angular margin between them have achieved remarkable success, notably in face recognition (Wang et al., 2018; Deng et al., 2019), large-scale dense retrieval tasks involving millions of classes (Zhu et al., 2021), as well as in diverse applications such as open-set recognition (Li et al., 2022), multi-label classification, and learning from imbalanced datasets (Li et al., 2024; Kang et al., 2021; Hayat et al., 2019; Ma et al., 2021; Shah et al., 2022). Moreover, uniformity, which encourages representations to be spread over the hypersphere, has been studied as an important property in representation learning and has been linked to downstream performance (Wang & Isola, 2020; Pu et al., 2022; Zhang et al., 2023). Accordingly, ensuring uniformity by providing additional margins on the hypersphere has been shown to be effective in representaion learning, making these approaches well-suited for the aforementioned applications.

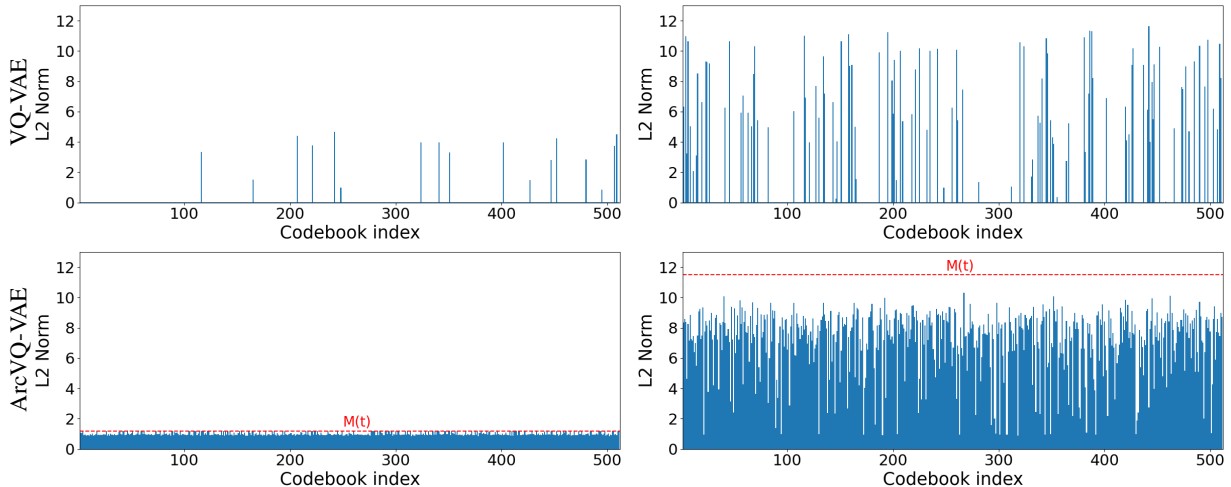

*Figure 2.* **Per index $\ell_2$-norms of codebook vectors.** The left column corresponds to the early stage of training, and the right column shows the distributions after substantial training. In VQ-VAE (top), only a small subset of codebook vectors exhibit large norms, indicating under-utilization and collapse. ArcVQ-VAE (bottom) maintains more uniformly bounded norms throughout training.

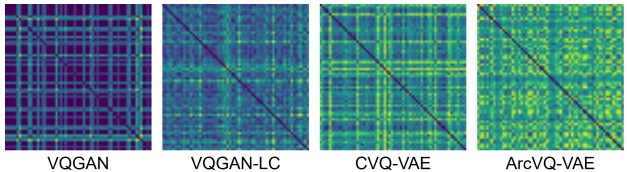

*Figure 3.* **Codebook pairwise $\ell_2$-distance matrices.** Each heatmap shows all pairwise Euclidean distances between the learned codebook vectors for each model. Brighter colors denote larger inter-codeword distances (greater separation).

## 3. Preliminary

Vector Quantized Variational Autoencoder (VQ-VAE) (Van Den Oord et al., 2017) is a discrete latent variable model that replaces continuous latent representations with vectors selected from a learnable codebook. Given an input image $x \in \mathbb{R}^{H \times W \times 3}$, an encoder $E(\cdot)$ maps it to a latent representation $z_e(x) \in \mathbb{R}^d$. This latent vector is then discretized via vector quantization by selecting the closest codebook entry $\mathbf{e}_k$ from a set of $K$ learnable embeddings $\mathcal{C} = \{\mathbf{e}_j\}_{j=1}^K$. The selection process is defined as follows:

$$z_q(x) = e_k, \quad \text{where} \quad k = \arg\min_j \|z_e(x) - e_j\|_2. \quad (1)$$

The selected quantized vector $z_q(x)$ is then passed to a decoder $D(\cdot)$, which reconstructs the original input $x$ as:

$$\hat{x} = D(z_q(x)) = D(\mathbf{q}(z_e(x))) = D(\mathbf{q}(E(x))), \quad (2)$$

where $\mathbf{q}(\cdot)$ is the quantization operator that replaces $z_e(x)$ with the nearest codebook vector.

Since the quantization operation is inherently non-differentiable, VQ-VAE employs the straight-through estimator (STE) (Bengio et al., 2013) to allow gradients to

flow during quantization. The overall objective is given by:

$$\mathcal{L}_{\text{VQ}} = \|x - \hat{x}\|_2^2 + \|z_q - \text{sg}(z_e)\|_2^2 + \beta\|\text{sg}(z_q) - z_e\|_2^2. \quad (3)$$

The first term minimizes the reconstruction error, the second term updates the codebook vectors toward the encoder outputs, and the third term encourages the encoder to commit more strongly to the selected codebook entries, with $\beta$ controlling the strength of this commitment. The stop-gradient operator $\text{sg}(\cdot)$ is used to block gradients from flowing through certain paths during backpropagation.

By learning discrete latent codes through vector quantization, VQ-VAE represents each data sample as a collection of codebook indices, such as a two-dimensional token grid for images. Based on these codebook index collections, generation can be performed by learning a prior distribution over the discrete tokens using either an autoregressive prior, such as PixelCNN (Van Den Oord et al., 2017) or Transformer (Esser et al., 2021), or a diffusion-based prior, such as a Latent Diffusion Model(LDM) (Rombach et al., 2022).

## 4. SAMP: Spherical Angular-Margin Prior

### 4.1. Codebook Analysis

Before the main discussion, we provide an analysis of the learned codebook vectors in terms of magnitudes and spatial distribution in the latent space. In conventional vector quantization-based models, we observe a distinct imbalance in the $\ell_2$-norms of the learned codebook vectors. Specifically, codebook vectors with high-utility (i.e. those frequently selected during training) tend to exhibit significantly larger norms, while low-utility vectors often retain near-zero norms, as shown in Figure 2. This discrepancy originates from the initialization and update process of the codebook.

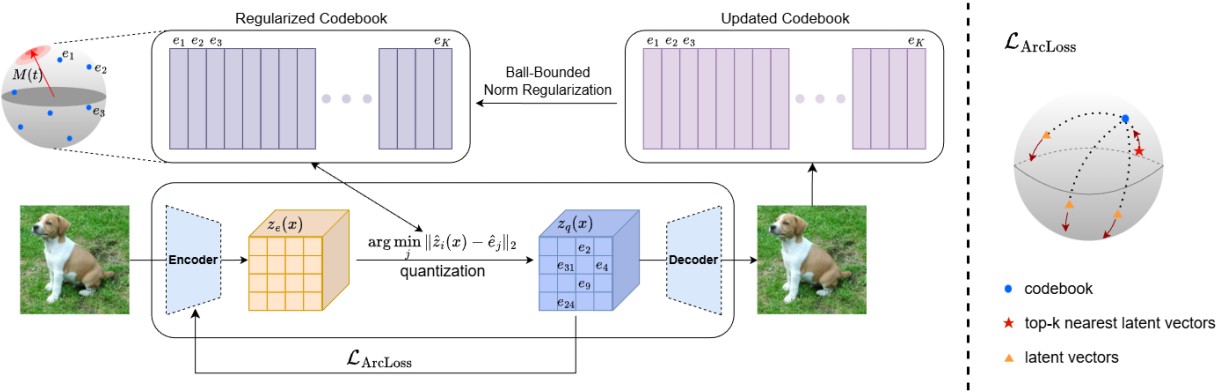

*Figure 4.* **The overall architecture of ArcVQ-VAE.** At each training step, the codebook vectors are rescaled using Ball-Bounded Norm Regularization to remain within a time-dependent Euclidean ball, enforcing controlled norm magnitudes. Simultaneously, the ArcLoss promotes angular dispersion among the latent vectors in the hyperspherical latent space while indirectly guiding the codebook vectors to form more discriminative and well-partitioned representations.

At the beginning of training, all codebook vectors are typically initialized near the origin. As training progresses, selected codebook vectors are updated in the direction of the encoded feature vectors $z_e(x)$, which generally have non-zero norms. Consequently, selected vectors gradually move away from the origin, accumulating larger magnitudes, whereas unselected vectors remain initial positions.

Over time, high-utility codes cluster around frequently occurring encoded features in the latent space (Zheng & Vedaldi, 2023), as shown in Figure 1. This concentration of frequently selected codes can reduce geometric diversity by limiting the effective coverage of different regions of the latent space. Figure 3 shows that the pairwise $\ell_2$-distance matrices reveal many codebook vectors lying close to one another, indicating that numerous entries coverage to nearly identical positions and thus become highly redundant. Moreover, being closer to the latent vectors, the high-utility codebook entries have a higher likelihood of being selected, leading to repeated usage and further strengthening their dominance. This leads to codebook collapse, where only a small subset of the entries is utilized while the rest remain inactive.

## 4.2. Ball-Bounded Norm Regularization

Thus, to suppress the excessive dominance of specific vectors in the latent space, we introduce a **Ball-Bounded Norm Regularization** that imposes an upper bound on the norm of each codebook vector. This regularization constrains all codebook vectors to reside within a time-dependent Euclidean ball in $\mathbb{R}^d$, thereby preventing certain codebook vectors from overly clustering around the latents and promoting more equitable competition among all codebook vectors with respect to their distance from the latent vectors. Specifically, we first $\ell_2$-normalize the codebook vectors

immediately after their random initialization:

$$\mathbf{e}_k^{(0)} \sim \ell_2(\text{Unif}(-1,1)^d), \quad \forall k \in \{1, \ldots, K\}, \quad (4)$$

which can be interpreted as placing the codebook vectors on the surface of a unit hypersphere in $\mathbb{R}^d$. As training progresses, the $\ell_2$-norm of each codebook vector is constrained not to exceed a time-dependent upper bound $M(t)$, which increases exponentially with the training step t:

$$M(t) = \exp(\alpha \cdot t), \quad (5)$$

$$\mathbf{e}_k^{(t)} \leftarrow \begin{cases} \dfrac{\mathbf{e}_k^{(t)}}{\|\mathbf{e}_k^{(t)}\|_2} \cdot M(t), & \text{if } \|\mathbf{e}_k(t)\|_2 > M(t) \\[2ex] \mathbf{e}_k(t), & \text{otherwise} \end{cases} \quad (6)$$

Here, $\alpha$ is a hyperparameter that controls the growth rate of the upper bound $M(t)$. We set $\alpha$ to a sufficiently small value (e.g. $10^{-5}$) so that $M(t)$ remains close to 1 for a substantial portion of the early training phase. This design gradually relaxes the norm constraint over time, allowing more flexibility in the magnitudes of the codebook vectors as training progresses. As a result, the model is encouraged to form a stable and balanced codebook distribution in the early stages, while enabling more expressive and adaptive vector representations in the later stages. Formally, the feasible region for the codebook at time step $t$ can be defined as a Euclidean ball of radius $M(t)$ in $\mathbb{R}^d$:

$$\mathcal{C}^{(t)} \subset \mathbb{B}_{M(t)}^d = \left\{ \mathbf{x} \in \mathbb{R}^d : \|\mathbf{x}\|_2 \leq M(t) \right\}. \quad (7)$$

## 4.3. ArcCosine Additive Margin Loss

Now, motivated by the geometric structure described in Section 4.2, where all codebook vectors are distributed within a time-dependent Euclidean ball, we introduce the concept

of **ArcCosine Additive Margin Loss (ArcLoss)** inspired from (Deng et al., 2019) into the VQ-VAE framework. This formulation let the latent vectors to be spread more evenly in the latent space by making them more angularly separable from each other. Consequently, the latent vectors capture more discriminative and fine-grained information, which enhances the quality of reconstruction. Simultaneously, this formulation induces each vector to occupy a distinct region of the latent space, thereby increasing the likelihood of associating with diverse codebook entries and improving overall codebook utilization.

First, we apply $\ell_2$-normalization to both the latent vectors and codebook vectors during quantization process. Specifically, given the encoder output $z_{e,i}(x) \in \mathbb{R}^d$ for the input image $x$ and the codebook vector $e_j$, we compute the normalized latent vector and codebook vector as follows:

$$\hat{z}_i(x) = \frac{z_{e,i}(x)}{\|z_{e,i}(x)\|_2}, \quad \hat{e}_j = \frac{e_j}{\|e_j\|_2}. \quad (8)$$

Here, $i \in \{1, \ldots, Bhw\}$ denotes the flattened index over all image tokens in the mini-batch, where $B$ is the batch size and $h$, $w$ represent the height and width of the latent feature map, respectively. By enforcing this normalization, the original codebook selection criterion in VQ-VAE, which selects the closest codebook entry based on the Euclidean distance, is now reinterpreted from the perspective of a unit hypersphere as a geodesic (i.e., angular) distance. The codebook entry is then selected by:

$$z_q(x) = e_k, \quad \text{where} \quad k = \arg\min_j \|\hat{z}_i(x) - \hat{e}_j\|_2$$
$$= \arg\max_j \hat{z}_i(x)^\top \hat{e}_j. \quad (9)$$

This formulation allows the normalized quantization rule to be interpreted as angular-similarity matching on the unit hypersphere.

Now, based on the spherical formulation of the latent space, we incorporate the ArcLoss into our VQ-VAE framework. Our goal is to explicitly encourage angular separation among the latent vectors and foster a more even dispersion throughout the hyperspherical latent space. To this end, we compute the angle between each normalized latent vector $\hat{z}_i(x)$ and a given normalized codebook entry $\hat{e}_j$ using the arccosine of their inner product,

$$\theta_{i,j} = \arccos(\hat{z}_i(x)^\top \hat{e}_j), \quad \theta_{i,j} \in [0, \pi]. \quad (10)$$

Based on these angular measurements, we define the ArcLoss $\mathcal{L}_\mathcal{A}$ to VQ-VAE model as:

$$\mathcal{L}_A = -\frac{1}{K} \sum_{j=1}^{K} \log \frac{\sum_{i \in \mathcal{N}_j^{(k)}} e^{s \cos(\theta_{i,j} + m)}}{\sum_{i \in \mathcal{N}_j^{(k)}} e^{s \cos(\theta_{i,j} + m)} + \sum_{i \notin \mathcal{N}_j^{(k)}}^{Bhw} e^{s \cos \theta_{i,j}}}. \quad (11)$$

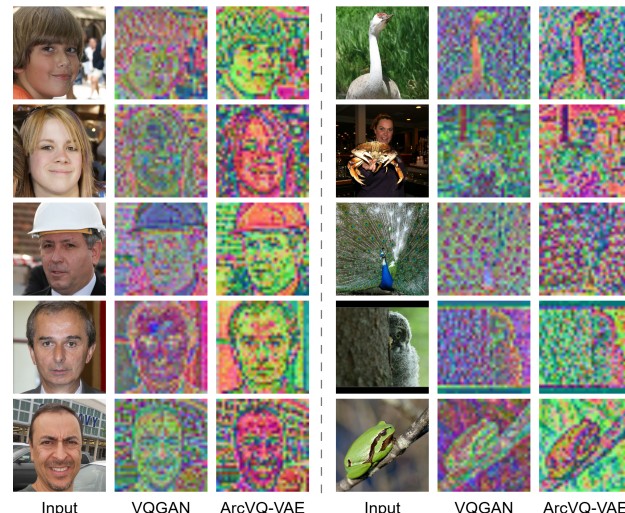

| Input | VQGAN | ArcVQ-VAE | Input | VQGAN | ArcVQ-VAE |

*Figure 5.* **Visualization of quantized latent maps.** For each input image, encoder features are quantized to codebook vectors and the assigned codebook vector at each spatial location is projected into the three RGB channels via PCA. ArcVQ-VAE exhibits more higher activation intensity and clearer contours.

where $K$ is the number of codebook entries, $s$ is a scaling factor, and $m$ is the additive angular margin. $\mathcal{N}_j^{(k)}$ denotes the index set of the top-$k$ latent tokens that are closest to $e_j$ on the unit hypersphere. This objective encourages the latent tokens in $\mathcal{N}_j^{(k)}$ to become angularly aligned with their associated codebook vector $e_j$, while pushing all remaining latents $\hat{z}_i(x)$ with $i \notin \mathcal{N}_j^{(k)}$ away from $e_j$ on the hypersphere. The additive angular margin $m$ enforces a tighter alignment for the positive pairs $(\hat{z}_i(x), e_j)$ with $i \in \mathcal{N}_j^{(k)}$ and enlarges the angular separation from negatives $(\hat{z}_i(x), e_j)$ with $i \notin \mathcal{N}_j^{(k)}$, leading to a more discriminative and well-partitioned hyperspherical latent space. Further theoretical details of the ArcLoss are provided in the appendix.

When computing the ArcLoss, we apply a stop-gradient operator to the codebook vectors $e_j$, so that gradients are backpropagated only to the encoder outputs. The angle in the loss is computed as:

$$\theta_{i,j} = \arccos\left(\hat{z}_i(x)^\top \text{sg}(\hat{e}_j)\right), \quad (12)$$

where $\text{sg}(\cdot)$ denotes the stop-gradient operator. This prevents ArcLoss from directly optimizing the codebook, which could otherwise collapse to a narrow, batch-driven distribution instead of maintaining global separability. Although ArcLoss does not update the codebook directly, it still affects it indirectly: as encoder outputs become more dispersed and align with different entries, the standard VQ loss pulls codebook vectors toward these spread features. Therefore, ArcLoss contributes positively to codebook diversity and utilization, allowing the codebook to more effectively cover the

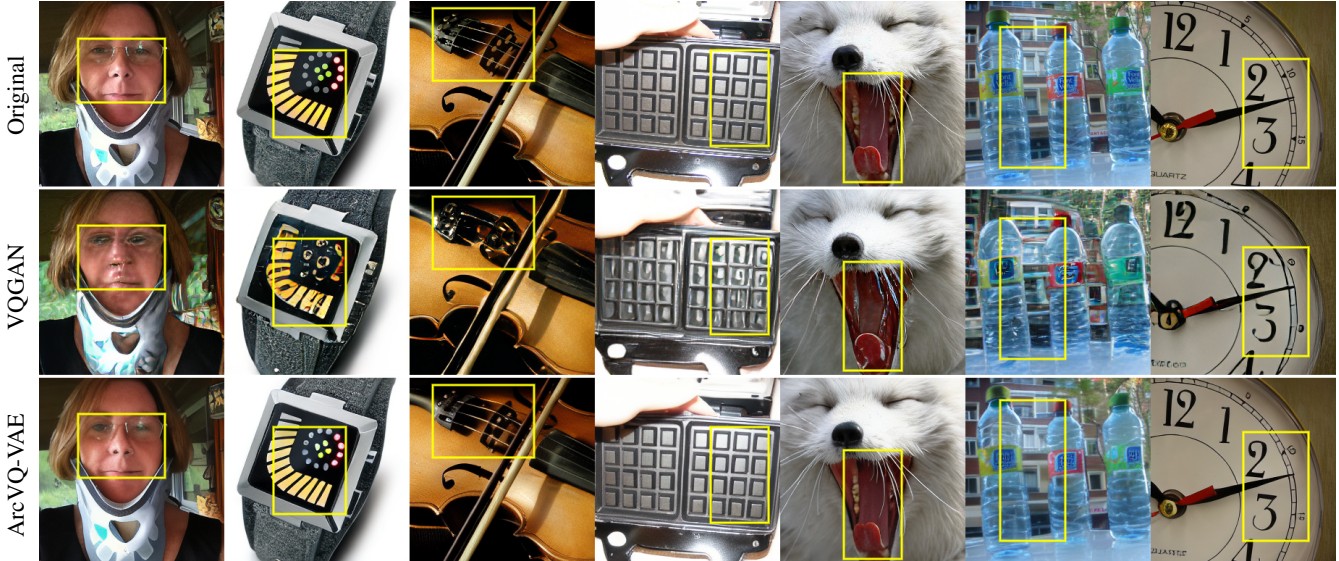

*Figure 6.* **Qualitative illustration of reconstruction quality.** Compared to the original images (top), our proposed ArcVQ-VAE (bottom) preserves local details more effectively than the baseline VQGAN (middle). The yellow-boxed regions highlight the improvements.

hyperspherical latent space. As shown in Figure 5, ArcVQ-VAE yields quantized latent maps with stronger activation intensity and clearer contours, indicating more discriminative and spatially structured codebook representations.

Finally, the overall objective function is given by:

$$\mathcal{L}_{\text{total}} = \mathcal{L}_{\text{VQ}} + \gamma(t)\mathcal{L}_{\text{A}}, \tag{13}$$

where $\gamma(t)$ is a decay factor that modulates the influence of the Arcloss in Eq. (11) during training. Specifically, it assigns a higher weight to the ArcLoss in the early stages of training to enforce strong angular constraints and gradually reduces its effect as training progresses, allowing the model to focus more on reconstruction accuracy.

## 5. Experiments

### 5.1. Experimental Setup

**Dataset and Metrics.** We primarily use the ImageNet-1K (Russakovsky et al., 2015) dataset for training. Also, to check the feasibility of the proposed method on smaller datasets, we use CIFAR-10 (Krizhevsky et al., 2009) and MNIST (Deng, 2012), following the previous comparison methods (Zheng & Vedaldi, 2023). For the image reconstruction, we evaluate performance using standard image quality metrics, including Structural Similarity Index Measure (SSIM), Peak Signal-to-Noise Ratio (PSNR), Learned Perceptual Image Patch Similarity (LPIPS) (Zhang et al., 2018), and Fréchet Inception Distance (FID) (Heusel et al., 2017). For image generation on ImageNet, we report FID, Inception Score (IS) (Salimans et al., 2016), precision, and recall (Kynkäänniemi et al., 2019).

**Baseline Methods.** We compare our proposed method against a range of existing vector-quantized generative models. For the small-scale datasets, MNIST and CIFAR-10, we evaluate our method against the following VQ-VAE-based baselines: VQ-VAE (Van Den Oord et al., 2017), HVQ-VAE (Williams et al., 2020), SQ-VAE (Takida et al., 2022), and CVQ-VAE (Zheng & Vedaldi, 2023). For ImageNet-1K dataset, we compare against representative VQGAN-based models, including the vanilla VQGAN (Esser et al., 2021), ViT-VQGAN (Yu et al., 2021), RQ-VAE (Lee et al., 2022), MoVQ (Zheng et al., 2022), SeQ-GAN (Gu et al., 2024), DF-VQGAN (Ni et al., 2023), CVQ-VAE (Zheng & Vedaldi, 2023) and VQGAN-LC (Zhu et al., 2024).

**Implementation Details.** To ensure fair comparisons, we use the same network architectures as the original VQGAN. We compress $256 \times 256$ input images with downsampling factors of $8\times$ and $16\times$, producing latent token maps of $32^2$ and $16^2$, respectively. For hyperparameter settings of the ArcLoss, we set the scaling factor $s = 10$, the additive angular margin $m = 0.1$, and select the top-$k = 3$ nearest latent tokens per codebook entry throughout all experiments. In addition, We set the decay factor as $\gamma(t) = \gamma_0 \cdot \exp(-\lambda t)$ with $\gamma_0 = 1.0$, where $\lambda = 5 \cdot 10^{-4}$ for MNIST and CIFAR-10, and $\lambda = 1 \cdot 10^{-4}$ for ImageNet.

For generation, we train an LDM (Rombach et al., 2022) as a generative prior on top of an ArcVQ-VAE trained with $32^2$ token maps. We follow the identical settings of the original LDM and use 250 sampling steps for ImageNet during inference. Additional implementation details are provided in the appendix.

*Table 1.* **Reconstruction results** on validation set of MNIST (10,000 images) and CIFAR10 (10,000 images).

| Method | $\ell_1$ loss ↓ | PSNR ↑ | SSIM ↑ | LPIPS ↓ | rFID ↓ |
|---|---|---|---|---|---|
| **MNIST** | | | | | |
| VQ-VAE | 0.0207 | 26.48 | 0.9777 | 0.0282 | 3.43 |
| HVQ-VAE | 0.0202 | 26.90 | 0.9790 | 0.0270 | 3.17 |
| SQ-VAE | 0.0197 | 27.49 | 0.9819 | 0.0256 | 3.05 |
| CVQ-VAE | 0.0180 | 27.87 | 0.9833 | 0.0222 | 1.80 |
| **ArcVQ-VAE** | **0.0178** | **28.01** | **0.9840** | **0.0217** | **1.68** |
| **CIFAR10** | | | | | |
| VQ-VAE | 0.0527 | 23.32 | 0.8595 | 0.2504 | 39.67 |
| HVQ-VAE | 0.0533 | 23.22 | 0.8553 | 0.2553 | 41.08 |
| SQ-VAE | 0.0482 | 24.07 | 0.8779 | 0.2333 | 37.92 |
| CVQ-VAE | 0.0448 | 24.72 | 0.8978 | 0.1883 | **24.73** |
| **ArcVQ-VAE** | **0.0445** | **24.78** | **0.8989** | **0.1857** | 26.91 |

*Table 2.* **Reconstruction results** on validation sets of ImageNet-1K (50,000 images). *S* denotes the token size, and *K* is the number of codevectors in the codebook.

| Method | Dataset | $S$ | $K$ | Usage | rFID↓ |
|---|---|---|---|---|---|
| VQGAN | | $16^2$ | 1024 | 44% | 7.94 |
| VQGAN-FC | | $16^2$ | 16384 | 11.2% | 4.29 |
| VQGAN-EMA | | $16^2$ | 16834 | 83.2% | 3.41 |
| Vit-VQGAN | | $32^2$ | 8192 | 96% | 1.28 |
| RQ-VAE | ImageNet | $8^2 \times 16$ | 2048 | - | 1.83 |
| MoVQ | | $16^2 \times 4$ | 1024 | 63% | 1.12 |
| SeQ-GAN | | $16^2$ | 1024 | 100% | 1.99 |
| CVQ-VAE | | $16^2$ | 1024 | 100% | 1.57 |
| CVQ-VAE | | $16^2 \times 4$ | 1024 | 100% | 1.20 |
| DF-VQGAN | | $32^2$ | 8192 | - | 1.38 |
| VQGAN-LC | | $32^2$ | 100000 | 99% | 1.29 |
| **ArcVQ-VAE** | | $16^2$ | 1024 | 99% | 1.63 |
| | | $32^2$ | 1024 | 88% | **0.95** |

### 5.2. Experimental Results

**Quantitative Results.** We report the quantitative reconstruction performance in Table 1 and Table 2. Our proposed method, ArcVQ-VAE, consistently achieves competitive or superior performance compared with other models. In particular, as shown in Table 2, our model delivers the largest gains at the $32^2$ token size. This suggests that our model can exploit a larger number of tokens more efficiently than prior approaches. Moreover, despite relying on a relatively small codebook of only $K = 1024$ entries, ArcVQ-VAE achieves superior reconstruction performance, indicating more effective codebook utilization during quantization and better empirical coverage of the latent space. Notably, these improvements are achieved without introducing any additional auxiliary modules or external models; rather, ArcVQ-VAE attains these gains solely by incorporating codebook regularization and an additional loss term, highlighting the effectiveness and simplicity of our design.

Quantitative results for generation performance are reported in Table 3. Our model exhibits superior results compared to existing approaches on ImageNet. This indicates that

*Table 3.* **Generation results** on ImageNet-1K. *S* denotes the token size, and *K* is the number of codevectors in the codebook.

| Method | $K$ | FID↓ | IS↑ | Prec↑ | rec↑ |
|---|---|---|---|---|---|
| VQGAN-FC (*LDM*) | 16384 | 9.78 | - | - | - |
| VQGAN-EMA (*LDM*) | 16384 | 9.13 | - | - | - |
| VQGAN-LC (*LDM*) | 100000 | 8.36 | 191.5 | 0.71 | 0.49 |
| RQVAE (*RQ-Transformer*) | 16384 | 7.55 | 134.0 | - | - |
| MoVQ (*MaskGIT*) | 1024 | 7.13 | 138.3 | 0.75 | 0.57 |
| CVQ-VAE (*LDM*) | 1024 | 6.87 | - | - | - |
| **ArcVQ-VAE** (*LDM*) | 1024 | **6.79** | 172.3 | 0.75 | 0.52 |

our approach, which covers more diverse regions of the latent space, is more effective when dealing with datasets containing a wide variety of categories. It is still notable that our model attains such performance while using only a codebook size of $K = 1024$.

**To summarize** the quantitative analysis, ArcVQ-VAE is competitive with strong VQ-based tokenizers and shows its most favorable results in the higher-resolution token setting and downstream generation. Furthermore, compared with vanilla VQGAN, SAMP also leads to noticeably improved codebook utility. Although our model does not reach 100% codebook usage, it still attains the best overall scores, suggesting that each codebook vectors are exploited more effectively during quantization. CVQ-VAE achieves 100% codebook usage and VQGAN-LC enlarges the codebook size to 100,000 while maintaining high utility, yet both fall short of the performance of ArcVQ-VAE. As shown in Figure 3, the pairwise $\ell_2$-distance matrix of ArcVQ-VAE is overall brighter and more homogeneous than those models, indicating that its codebook covers a broader and more diverse region of the latent space, thereby endowing each codebook vector with higher representation quality and ultimately yielding superior reconstruction and generation performance. This observation is also consistent with previous representation learning studies that emphasize uniformity (Wang & Isola, 2020; Yao et al., 2025).

**Qualitative Results.** We present qualitative comparisons of reconstruction quality in Figure 6. As illustrated in the figure, our method more effectively preserves fine-grained visual details, such as facial components, object contours, small text, and edge patterns. Regions marked with yellow boxes highlight areas where ArcVQ-VAE achieves clearer and more faithful reconstructions than VQGAN. In particular, VQGAN often under-allocates codebook capacity to surrounding structures and background context, which leads to visible artifacts outside the main objects. By contrast, ArcVQ-VAE allocates codebook capacity more evenly across foreground and background, which preserves fine structures, and keeps reconstructions sharp and faithful even in cluttered or low-contrast regions.

Figure 7 shows samples generated by an LDM equipped

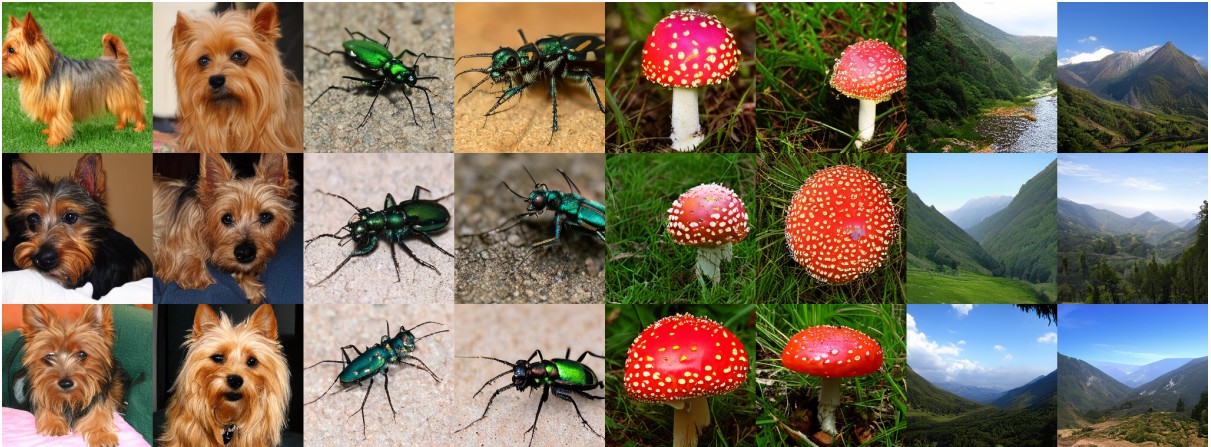

*Figure 7.* **Qualitative illustration of generation quality on ImageNet.** The images are generated by an LDM equipped with ArcVQ-VAE tokenizer under class-conditional settings, using $32 \times 32$ token maps and 250 sampling steps.

*Table 4.* **Ablation Study** on various settings of top-$k$, $m$, and $s$.

| top-$k$ | MNIST | | | CIFAR10 | | |
|---|---|---|---|---|---|---|
| | PSNR↑ | SSIM↑ | rFID↓ | PSNR↑ | SSIM↑ | rFID↓ |
| 1 | 28.07 | 0.9841 | 1.70 | 24.70 | 0.8990 | 27.08 |
| 3 | 28.01 | 0.9840 | 1.68 | 24.78 | 0.8989 | 26.91 |
| 8 | 28.19 | 0.9844 | 1.72 | 24.69 | 0.8966 | 27.02 |

| $m$ | MNIST | | | CIFAR10 | | |
|---|---|---|---|---|---|---|
| | PSNR↑ | SSIM↑ | rFID↓ | PSNR↑ | SSIM↑ | rFID↓ |
| 0.1 | 28.01 | 0.9840 | 1.68 | 24.78 | 0.8989 | 26.91 |
| 0.5 | 28.05 | 0.9840 | 1.77 | 24.75 | 0.8980 | 27.93 |
| 1.0 | 28.03 | 0.9844 | 1.72 | 24.68 | 0.8965 | 28.52 |

| $s$ | MNIST | | | CIFAR10 | | |
|---|---|---|---|---|---|---|
| | PSNR↑ | SSIM↑ | rFID↓ | PSNR↑ | SSIM↑ | rFID↓ |
| 5 | 28.03 | 0.9839 | 1.65 | 24.65 | 0.8980 | 26.83 |
| 10 | 28.01 | 0.9840 | 1.68 | 24.78 | 0.8989 | 26.91 |
| 20 | 27.92 | 0.9835 | 1.87 | 23.95 | 0.8970 | 28.90 |

*Table 5.* **Ablation study** on various codebook dimensions $D$. Experiments are conducted on the FFHQ dataset. $\ell_2$-norm is the average over codebook vectors used during reconstruction.

| $D$ | $\ell_2$-norm | $K$ | Usage | PSNR↑ | SSIM↑ | rFID↓ |
|---|---|---|---|---|---|---|
| 4 | 1.81 | 1024 | **98%** | 25.69 | 0.8021 | 2.71 |
| 8 | 1.95 | 1024 | 87% | **26.71** | 0.8124 | 2.24 |
| 16 | 2.64 | 1024 | 78% | 26.28 | **0.8128** | **2.14** |

*Table 6.* **Component ablations.** Starting from a vanilla VQ-VAE baseline, we sequentially add BBNR(Ball-Bounded Norm Regularization) and ArcLoss.

| Method | MNIST | | | CIFAR10 | | |
|---|---|---|---|---|---|---|
| | PSNR↑ | SSIM↑ | rFID↓ | PSNR↑ | SSIM↑ | rFID↓ |
| VQ-VAE | 26.48 | 0.9777 | 3.43 | 23.32 | 0.8595 | 39.67 |
| + BBNR | 27.39 | 0.9814 | 2.71 | 23.97 | 0.8774 | 35.18 |
| + ArcLoss | 28.01 | 0.9840 | 1.68 | 24.78 | 0.8989 | 26.91 |

with ArcVQ-VAE tokenizer across diverse object and scene categories. The model produces fine object details (e.g., fur texture, cap spot patterns, mountain shading), while faithfully preserving local background details (e.g., blanket folds, soil textures, blades of grass), suggesting improved overall representation qaulity.

## 5.3. Ablation Study

We conduct a series of ablations to evaluate sensitivity to hyperparameter choices and the contribution of each component. First, we vary the top-$k$ selection ,the additive angular margin $m$, and the scaling factor $s$. As demonstrated in Table 4, performance remains largely invariant across various settings, indicating that our proposed ArcVQ-VAE is robust to hyperparameter choices.

Second, we vary the dimensionality of the codebook vec-

tors on FFHQ (Karras et al., 2019) dataset. As shown in Table 5, lower dimensional codebooks achieve higher codebook usage, whereas higher-dimensional codebook yield lower rFID. We attribute this behavior to a trade-off between geometric fidelity and representational capacity. When the dimensionality is small, the codebook vectors exhibit smaller $\ell_2$-norms that better respect the underlying spherical geometry, leading to higher codebook utility, while larger dimensions enable richer feature encoding, which improves rFID performance.

Finally, we evaluate each core component. As shown in Table 6, introducing Ball-Bounded Norm Regularization to the baseline yields consistent improvements, and incorporating ArcLoss further enhances performance. The larger gains from ArcLoss indicate that encouraging angular separation in the latent space is especially beneficial, thereby corroborating our analysis.

# 6. Conclusion

In this paper, we introduced ArcVQ-VAE, a vector-quantization framework that imposes a spherical angular-margin prior among codebook vectors by combining Ball-Bounded Norm Regularization with an ArcCosine Additive-Margin loss. This geometry encourages discriminative, uniformly dispersed latents inside a Euclidean ball, thereby enabling the codebook to cover a broader range of latent patterns, enhancing the overall representation quality. Experiments demonstrate consistent gains over baselines in both reconstruction and generative fidelity, alongside more semantically efficient codebook use. We hope this framework inspires and informs future geometry-aware advances in vector quantization. Despite achieving state-of-the-art performance, a promising direction for future work is to integrate our approach with recent advances in VQ-VAE methods.

# Acknowledgements

This work was supported by the Institute of Information & Communications Technology Planning & Evaluation (IITP) grant funded by the Korea government (MSIT) [RS-2021-II211341, Artificial Intelligence Graduate School Program (Chung-Ang University) and RS-2022-II220124, Development of Artificial Intelligence Technology for Self-Improving Competency-Aware Learning Capabilities]. SNUAILAB, corp, supports this work.

# Impact Statement

This work improves vector-quantized autoencoders by promoting more uniform and stable codebook usage on a hypersphere, enabling better reconstructions and more reliable discrete representations without substantial architectural changes. These gains can benefit downstream applications such as image compression, efficient representation learning, and scalable generative modeling, reducing wasted capacity and making more effective use of limited codebook budgets. The method does not introduce new data or collect personal information, and we do not anticipate ethical risks beyond those already associated with models that learn discrete image representations and support generative pipelines. The primary broader impact is improved efficiency and practicality: stronger codebook representations can stabilize training, preserve or improve reconstruction quality, and reduce compute and iteration costs for researchers and practitioners.

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

## A. Additive Angular Margin Loss

Softmax-based classification loss is widely used due to its simplicity and effectiveness. However, the conventional softmax loss does not explicitly optimize the embedding space for intra-class compactness and inter-class separability, which can lead to suboptimal performance, especially in scenarios involving large intra-class variations such as changes in pose or age. To address these limitations, Ar-cFace (Deng et al., 2019) introduces an additive angular margin penalty that enhances the discriminative power of deep features. The original softmax loss is formulated as follows:

$$\mathcal{L}_{\text{softmax}} = -\log \left( \frac{e^{W_{y_j}^\top x_j + b_{y_j}}}{\sum_{i=1}^{N} e^{W_i^\top x_j + b_i}} \right), \qquad (14)$$

where $x_j \in \mathbb{R}^d$ is the embedding of the $j$-th sample belonging to class $y_j$, $W_i$ is the $i$-th column of the weight matrix $W \in \mathbb{R}^{\times \mathbb{N}}$, $b_j \in \mathbb{R}^{\mathbb{N}}$ is the bias term, and $N$ is the total number of classes.

For better geometric interpretation and to isolate the angular component of similarity, the bias terms are removed and both the weights and features are $\ell_2$-normalized. Specifically, the logit is rewritten using the cosine similarity:

$$W_i^\top x_j = \|W_i\| \cdot \|x_j\| \cdot \cos\theta_i \to s \cdot \cos\theta_i, \qquad (15)$$

where $\theta_i$ is the angle between $W_i$ and $x_j$, and $s$ is a fixed scaling factor applied after normalization. This reformulation ensures that all embedding vectors lie on the surface of a hypersphere, with classification depending solely on angular similarity. Further, as introduced in Deng et al. (2019), we can modify the formulation by introducing an additive angular margin $m$ to the target logit:

$$\mathcal{L}_{\text{ArcFace}} = -\log \frac{e^{s\cos(\theta_{y_j}+m)}}{e^{s\cos(\theta_{y_j}+m)} + \sum_{i=1,i\neq y_j}^{N} e^{s\cos\theta_i}}, \qquad (16)$$

which enforces a stricter decision boundary by increasing the angular separation between classes. This angular margin encourages features from the same class to be closer together on the hypersphere while pushing features of different classes further apart, leading to improved performance.

Building upon this angular margin formulation from Ar-cFace, we design an analogous loss for vector-quantized generative models. For a given normalized latent vector $\hat{z}_i(x)$ and normalized codebook entry $\hat{e}_j$, we define the Ar-cLoss $\mathcal{L}_\mathcal{A}$ to VQ-VAE model as:

$$\theta_{i,j} = \arccos\left( \hat{z}_i(x)^\top \hat{e}_j \right), \qquad (17)$$

$$\mathcal{L}_{\text{A}} = -\frac{1}{K}\sum_{j=1}^{K} \log \frac{\sum_{i\in\mathcal{N}_j^{(k)}} e^{s\cos(\theta_{i,j}+m)}}{\sum_{i\in\mathcal{N}_j^{(k)}} e^{s\cos(\theta_{i,j}+m)} + \sum_{i\notin\mathcal{N}_j^{(k)}}^{Bhw} e^{s\cos\theta_{i,j}}}. \qquad (18)$$

where $K$ is the number of codebook entries, and $\mathcal{N}_j^{(k)}$ denotes the index set of the top-$k$ latent tokens that are closest to $e_j$ on the unit hypersphere.

From a probabilistic viewpoint, for each codebook entry $j$, the inner term of Eq. (18) can be interpreted as the negative log-likelihood of a binary softmax classifier that distinguishes the positive latent tokens $\mathcal{N}_j^{(k)}$ from the complementary negative set $\{1, \ldots, Bhw\} \setminus \mathcal{N}_j^{(k)}$. The logits corresponding to the positive indices $i \in \mathcal{N}_j^{(k)}$ are given by $s\cos(\theta_{i,j} + m)$, while those for negative indices $i \notin \mathcal{N}_j^{(k)}$ are given by $s\cos\theta_{i,j}$. Averaging this binary softmax objective over all $K$ codebook entries yields ArcLoss, which can be seen as a codebook-centric contrastive objective where each codebook vector $e_j$ serves as an anchor, the top-$k$ neighbors from the positive set, and all remaining latent tokens act as negatives.

Intuitively, ArcLoss inherits the additive angular margin mechanism of ArcFace in a codebook latent setting. The margin $m > 0$ does not change the ideal alignment point for positive pairs, which remains $\theta_{i,j} = 0$. Instead, it makes the positive matching criterion stricter. Since $\theta_{i,j} \in [0, \pi]$ and $\cos(\theta)$ is monotonically decreasing on this interval, replacing $\cos\theta_{i,j}$ with $\cos(\theta_{i,j} + m)$ lowers the positive logit unless the positive pair achieves a smaller angle. Thus, minimizing ArcLoss encourages the top-$k$ latent tokens to align more tightly with their associated codebook vector, while the remaining latent tokens are treated as negatives and separated through the softmax objective. As a result, each codebook entry tends to form a compact angular neighborhood of nearby latent tokens and maintain clearer angular separation from non-neighbor tokens, leading to a more structured partition of the hyperspherical latent space and more balanced codebook utilization.

## B. Implementation Details

For the small-scale datasets, MNIST and CIFAR-10, we conduct experiments using the officially released VQ-VAE implementation. For ImageNet-1K, we adopt the officially released VQGAN and LDM architectures, and train all models on a single NVIDIA H100 (80GB) GPU using the Adam optimizer.

The hyperparameters used in our experiments are summarized in Table 7 and Table 8. Across all settings, all models share the same ArcLoss hyperparameters, the scaling factor $s$, angular margin $m$, top-$k$, initial weight $\gamma_0$, and decay rate

*Table 7.* Implementation Details for VQ-VAE and VQGAN

| Dataset | MNIST | CIFAR-10 | ImageNet |
|---|---|---|---|
| Input Size | $28 \times 28$ | $32 \times 32$ | $256 \times 256$ |
| Downsampling | $4\times$ | $4\times$ | $8\times$ |
| Dimension | 64 | 64 | 16 |
| Codebook Size | 512 | 512 | 1024 |
| $\alpha$ | 3e-4 | 3e-4 | 1e-5 |
| $s$ | 10 | 10 | 10 |
| $m$ | 0.1 | 0.1 | 0.1 |
| top-$k$ | 3 | 3 | 3 |
| $\gamma_0$ | 1.0 | 1.0 | 1.0 |
| $\lambda$ | 5e-4 | 5e-4 | 1e-4 |
| Batch Size | 1024 | 1024 | 30 |
| epochs | 500 | 500 | 15 |
| Learning Rate | 3e-4 | 3e-4 | 4.5e-6 |

*Table 8.* Implementation Details for LDM

| Dataset | ImageNet |
|---|---|
| Input Size | $256 \times 256$ |
| $z$-shape | $32 \times 32 \times 16$ |
| Codebook Size | 1024 |
| Noise Schedule | linear |
| Channels | 256 |
| Depth | 2 |
| Attention Resolution | 32, 16, 8 |
| Head Channels | 32 |
| Batch Size | 64 |
| Training Iterations | 1.2M |
| Learning Rate | 1.0e-5 |
| DDIM steps | 250 |

$\lambda$. For the Ball-Bounded Norm Regularization, we adjust the $\alpha$ according to the target number of training iterations, since $\alpha$ controls the growth of the upper bound on the norms of the codebook vectors. A smaller value of $\alpha$ improves the stability of codebook utilization in the early phase of training, but if $\alpha$ is set too small, the norm constraint remains overly restrictive throughout training, limiting the flexibility of the codebook vector norms and potentially degrading performance.

# C. Operational Meaning of Codebook Imbalance

We emphasize that the term geometric imbalance is used operationally in this paper to describe empirical geometric patterns observed in learned codebooks, including norm imbalance and spatial concentration. We do not claim that all codebook vectors should have identical norms or be perfectly uniformly distributed in the latent space, nor that such a state is theoretically optimal. Rather, norm imbalance refers to the empirical skew observed in Figure 2: frequently selected codebook vectors accumulate larger norms through repeated updates, whereas rarely selected or unused vectors tend to remain close to their initialization with much smaller norms.

This imbalance naturally arises from the update dynamics of vector quantization. Codebook vectors selected by nearest-neighbor assignment are repeatedly updated toward encoder outputs with non-zero norms, while rarely selected vectors receive few effective updates. Therefore, the observed norm imbalance should be understood as an empirical signature of uneven codebook utilization, not as evidence that equal norms are universally desirable.

Our goal is not to enforce identical norms across all codebook entries or to impose a perfectly uniform codebook distribution. Ball-Bounded Norm Regularization only imposes a time-dependent upper bound on the codebook norm, preventing a small subset of frequently selected vectors from growing excessively during early training. Thus, the proposed constraint mitigates early dominance and improves effective latent-space coverage, rather than imposing an artificial notion of perfect geometric balance.

# D. Additional Results

## D.1. Comparison with Hyperspherical VAE

To further examine the effect of using a VQ-based discrete latent formulation, we additionally compare VAE (Kingma et al., 2013), hyperspherical VAE (Davidson et al., 2018), standard VQ-VAE, and ArcVQ-VAE under matched reconstruction settings. The hyperspherical VAE uses the same encoder and decoder architecture as the VAE baseline, but replaces the Gaussian latent variable with a hyperspherical latent variable based on a von Mises-Fisher posterior and a hyperspherical uniform prior. This comparison allows us to distinguish the effect of imposing hyperspherical geometry in a continuous latent space from the effect of learning a discrete codebook-based representation.

As shown in Table 9, hyperspherical VAE improves over the standard VAE in several metrics, indicating that hyperspherical latent regularization can be beneficial. However, it remains substantially worse than the VQ-based models on MNIST and CIFAR-10. This suggests that, in our setting, a discrete codebook-based tokenizer provides a stronger reconstruction and representation structure than a continuous hyperspherical latent-space model. ArcVQ-VAE improves over the standard VQ-VAE by imposing spherical angular-

*Table 9.* Reconstruction comparison with hyperspherical VAE. Hyperspherical VAE improves over the standard VAE, while VQ-VAE and ArcVQ-VAE achieve stronger reconstruction performance.

| MNIST | | | | |
|---|---|---|---|---|
| Method | PSNR ↑ | SSIM ↑ | LPIPS ↓ | rFID ↓ |
| VAE | 18.20 | 0.8505 | 0.0918 | 15.34 |
| hyperspherical VAE | 18.71 | 0.8720 | 0.0913 | 12.87 |
| VQ-VAE | 26.48 | 0.9777 | 0.0282 | 3.43 |
| **ArcVQ-VAE** | **28.13** | **0.9843** | **0.0213** | **1.59** |
| CIFAR-10 | | | | |
| Method | PSNR ↑ | SSIM ↑ | LPIPS ↓ | rFID ↓ |
| VAE | 17.45 | 0.5609 | 0.5120 | 97.76 |
| hyperspherical VAE | 17.48 | 0.6830 | 0.5459 | 89.51 |
| VQ-VAE | 23.32 | 0.8595 | 0.2504 | 39.67 |
| **ArcVQ-VAE** | **24.71** | **0.8976** | **0.1928** | **27.17** |

margin regularization on the discrete codebook geometry. Therefore, the proposed method should be interpreted not as a replacement for general hyperspherical latent-variable modeling, but as a geometry-aware regularization strategy specifically designed for VQ-based discrete tokenizers.

### D.2. Post-hoc Codebook Reduction Analysis

To further analyze the relationship between codebook size, codebook usage, and reconstruction quality, we conduct a post-hoc codebook reduction experiment on VQGAN-LC. Starting from a pretrained VQGAN-LC model with 100,000 codebook entries, we reduce the codebook size by applying $k$-means clustering to the pretrained codebook vectors and then re-evaluate reconstruction performance without retraining the tokenizer. We denote this reduced-codebook variant as VQGAN-LC (R).

*Table 10.* Post-hoc codebook reduction results on VQGAN-LC. (R) denotes the reduced-codebook variant obtained by applying $k$-means clustering to the pretrained codebook vectors.

| Method | Codebook Size | Usage | rFID ↓ |
|---|---|---|---|
| VQGAN-LC | 100000 | 99.5% | 1.13 |
| VQGAN-LC (R) | 50000 | 99.5% | 1.13 |
| VQGAN-LC (R) | 16384 | 99.7% | 1.13 |
| VQGAN-LC (R) | 8192 | 99.9% | 1.14 |
| VQGAN-LC (R) | 4096 | 100% | 1.26 |
| VQGAN-LC (R) | 1024 | 100% | 2.40 |
| VQGAN-LC (R) | 32 | 100% | 79.2 |

As shown in Table 10, reconstruction quality remains nearly unchanged when the VQGAN-LC codebook is reduced from 100,000 to 8,192 entries, while codebook usage remains close to 100%. However, further reducing the codebook size leads to a clear degradation in rFID, even though the measured usage remains 100%. This suggests that high usage alone does not fully determine tokenizer quality, since

*Table 11.* Results of cosine-similarity matching on CIFAR-10. Cosine-similarity matching alone yields only marginal changes compared with vanilla VQ-VAE, whereas ArcVQ-VAE provides clear improvements across all metrics.

| Method | PSNR ↑ | SSIM ↑ | LPIPS ↓ | rFID ↓ |
|---|---|---|---|---|
| VQ-VAE | 23.32 | 0.8595 | 0.2504 | 39.67 |
| VQ-VAE(cosine-sim.) | 23.37 | 0.8607 | 0.2466 | 39.04 |
| **ArcVQ-VAE** | **24.71** | **0.8976** | **0.1928** | **27.17** |

the number of available entries and the redundancy structure of the codebook also play important roles. Accordingly, we interpret ArcVQ-VAE as a tokenizer that improves codebook geometry and downstream quality under a relatively small codebook budget, rather than as a method that simply maximizes codebook usage.

### D.3. Effect of Cosine Similarity Matching

ArcVQ-VAE performs quantization using $\ell_2$-normalized latent vectors and codebook vectors, which makes nearest-neighbor search equivalent to cosine-similarity matching as described in Eq. 9. To verify that the improvement does not come merely from cosine-based quantization, we compare vanilla VQ-VAE, VQ-VAE with cosine-similarity matching, and ArcVQ-VAE under the same CIFAR-10 setting. The cosine-similarity baseline normalizes both encoder outputs and codebook vectors only during quantization, without Ball-Bounded Norm Regularization or the additive angular-margin objective.

As shown in Table 11, cosine-similarity matching alone provides only marginal improvements over vanilla VQ-VAE. In contrast, ArcVQ-VAE substantially improves reconstruction quality and perceptual metrics under the same setting. This indicates that the empirical gains of ArcVQ-VAE do not come simply from using cosine similarity for quantization, but from the combination of Ball-Bounded Norm Regularization and the additive angular-margin objective. The angular margin makes normalized cosine-based matching stricter by assigning positive pairs logits of the form $\cos(\theta_{i,j} + m)$, while negative pairs retain $\cos(\theta_{i,j})$. Since cosine is monotonically decreasing on $[0, \pi]$ and $m > 0$ lowers the positive logits unless a stronger angular alignment is achieved, the model is encouraged to align latent tokens more closely with their assigned codebook entries while separating them from others.

### D.4. Effect of Norm-Bound Schedule

To further assess the role of the Ball-Bounded Norm Regularization schedule, we compare the original ArcVQ-VAE with a fixed-bound variant that keeps the ArcLoss unchanged but fixes the norm bound to $M(t) = 1$ for all training iterations. This variant allows us to isolate the

*Table 12.* Reconstruction results under different norm-bound schedules. The fixed-bound variant with $M(t) = 1$ improves over vanilla VQ-VAE, while the original time-dependent schedule achieves the best performance.

| MNIST | | | | |
|---|---|---|---|---|
| Method | PSNR $\uparrow$ | SSIM $\uparrow$ | LPIPS $\downarrow$ | rFID $\downarrow$ |
| VQ-VAE | 26.48 | 0.9777 | 0.0282 | 3.43 |
| ArcVQ-VAE ($M(t) = 1$) | 27.47 | 0.9786 | 0.0257 | 1.97 |
| **ArcVQ-VAE** | **28.13** | **0.9843** | **0.0213** | **1.59** |
| CIFAR-10 | | | | |
| Method | PSNR $\uparrow$ | SSIM $\uparrow$ | LPIPS $\downarrow$ | rFID $\downarrow$ |
| VQ-VAE | 23.32 | 0.8595 | 0.2504 | 39.67 |
| ArcVQ-VAE ($M(t) = 1$) | 24.01 | 0.8797 | 0.2112 | 30.70 |
| **ArcVQ-VAE** | **24.71** | **0.8976** | **0.1928** | **27.17** |

effect of the time-dependent relaxation schedule from the effect of imposing a norm constraint itself.

As shown in Table 12, fixing $M(t) = 1$ throughout training still improves over vanilla VQ-VAE. This indicates that constraining the codebook norm itself is beneficial. However, the original time-dependent schedule, $M(t) = \exp(\alpha t)$, consistently outperforms the fixed-bound variant. These results suggest that a constant bound can stabilize early training but may remain overly restrictive in later stages. By contrast, the time-dependent schedule provides a stronger constraint in the early phase while gradually relaxing the feasible region of the codebook vectors, allowing greater flexibility as training progresses.

### D.5. Transfer to an Autoregressive PixelCNN Prior

To examine whether the benefit of ArcVQ-VAE is specific to the Latent Diffusion Model (LDM) prior used in our ImageNet generation experiments, we additionally evaluate the learned tokenizer with an autoregressive PixelCNN prior. This experiment compares a vanilla VQ-VAE tokenizer and an ArcVQ-VAE tokenizer under the same downstream PixelCNN architecture and training protocol on CIFAR-10. Since ArcVQ-VAE modifies only the tokenizer and not the downstream sampler, this comparison helps assess whether the learned discrete representations are also beneficial for a different class of generative priors.

As shown in Table 13, replacing the vanilla VQ-VAE tokenizer with the ArcVQ-VAE tokenizer improves the downstream FID from 57.36 to 43.13 under the same PixelCNN prior. This result suggests that the benefit of ArcVQ-VAE is not limited to the LDM-based generation setup. Rather, the proposed tokenizer can provide discrete representations that are useful for different classes of downstream generative priors. We emphasize that this experiment is intended to support tokenizer-side transferability across multiple posterior or prior modeling choices, rather than to claim superiority for any particular sampler.

*Table 13.* Generation results with a PixelCNN prior on CIFAR-10. Replacing the vanilla VQ-VAE tokenizer with the ArcVQ-VAE tokenizer improves FID under the same PixelCNN setup.

| Method | Token Size | Codebook Size | FID $\downarrow$ |
|---|---|---|---|
| PixelCNN(VQ-VAE) | 64 | 512 | 57.36 |
| **PixelCNN(ArcVQ-VAE)** | **64** | **512** | **43.13** |

## E. Visualizations

We present qualitative visualization of images at a resolution of $256 \times 256$, generated by an LDM trained on our ArcVQ-VAE tokenizer. Figure 8 show class-conditional ImageNet samples obtained using $32 \times 32$ latent tokens, a classifier-free guidance scale of 1.4, and 250 DDIM sampling steps.

## F. Limitations and Future Work

Although ArcVQ-VAE shows favorable empirical performance, several limitations remain. First, some design choices are heuristic. The top-$k$ selection in Eq. 11 and the loss-weight schedule $\gamma(t)$ in Eq. 13 are chosen as practical defaults rather than theoretically optimal settings. Table 4 shows that the method is relatively robust to moderate changes in $k$, $m$, and $s$, but a more exhaustive study of alternative schedules, including warm-up, constant, oscillating, and adaptive profiles that respond to training dynamics, is left for future work.

Second, ArcVQ-VAE improves the tokenizer but does not remove the need for a downstream generative prior. Although the codebook can be viewed as a dictionary-like latent structure, it does not by itself define a distribution over codebook index grids. Therefore, as in standard VQ-based generative modeling, a prior over discrete tokens must still be learned. Our PixelCNN experiment suggests that the benefit is not limited to the LDM-based setup, but whether improved codebook geometry can simplify downstream prior learning remains future work.

Finally, our current claims are mainly empirical and geometric. We provide evidence for improved codebook usage, reduced redundancy, spatially structured quantized maps, and favorable reconstruction or generation performance, but we do not provide a direct semantic benchmark for codebook entries or a rigorous STE gradient-level analysis. Future work could address these points through more direct semantic evaluation and deeper theoretical analysis.

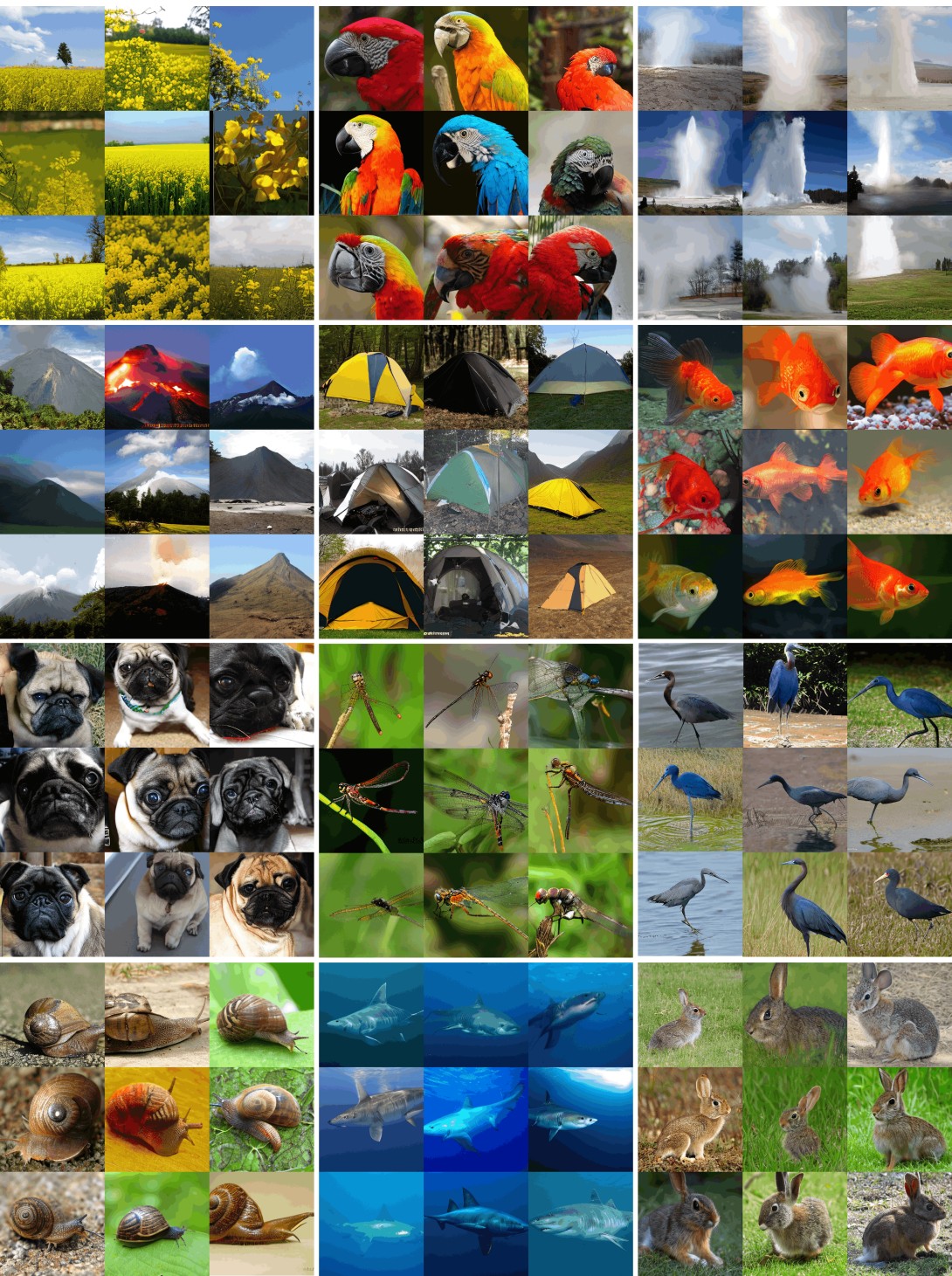

*Figure 8.* **Class-conditional ImageNet-1K samples** at a resolution of $256 \times 256$ generated by LDM trained on the ArcVQ-VAE tokenizer, using $32 \times 32$ latent tokens, a classifier-free guidance scale of 1.4, and 250 DDIM sampling steps.

