# OpenReview forum: "ArcVQ-VAE: A Spherical Vector Quantization Framework with ArcCosine Additive Margin"
_ICML.cc/2026/Conference — ICML 2026 regular_

### Official Review · Reviewer_89GT · 2026-02-16

**Soundness:** 3
**Presentation:** 3
**Significance:** 3
**Originality:** 3
**Overall Recommendation:** 3
**Confidence:** 4

**Summary:**

The paper is based on the VQ-VAE method. It proposes to introduice a weak normalisation of the dictionary vectors estimated along the training to enforce the use of the majority of these vectors. The authors also include a ArcCosine Additive Margin Loss to enforce the vectors to spread over the unit hyper-sphere.

**Compliance With Llm Reviewing Policy:**

Affirmed.

**Final Justification:**

Still not convinced by the arguments.

**Key Questions For Authors:**

1- What is the impact on LDM vs other posterior resampling models?
2- Can you use the "dictionary learning" like representation in the latent space to avoid training a posterior sampler? If yes, how?

**Limitations:**

See weaknesses.

**Strengths And Weaknesses:**

The paper technically sounds and the objective is clear and well acheived.
The experiments are showing the effect of the added elements to the standard VQ-VAE method.

However, it would have been interesting to compare this method to standard VQ-VAE rather than VQ-GAN and also compare to methods which assume a representation of the data as elements on the hypersphere (like hyperspherical VAE like methods). Although different as they are not using the dictionary representation in the latent space, the idea is quite close and a comparison would have been valuable. This would have also help understand the interest in this Vector Quantized framwork wrt the general latent space case.

I would also have appreciated to see the effect of forcing M(t) = 1 for all iterations or changing its decreasing shape (for example a warm up phase before decrease or an oscilating decreasing profile).

---

> ### Author Rebuttal · Authors · 2026-03-31
>
> Thank you for your thoughtful and constructive feedback. We appreciate your positive assessment of the technical soundness and clarity of our work. We respond to your concerns as follows.
>
> >W1. Comparison to standard VQ-VAE and to hyperspherical latent-space methods
>
> Thank you for this important comment. We agree that both comparisons are relevant.
> First, our submission already includes a direct comparison to standard VQ-VAE in Tables 1 and 5. In particular, Table 5 shows that both components improve upon the standard formulation. We agree, however, that this point was not emphasized clearly enough, since the large-scale tables focus on stronger tokenizer baselines such as VQGAN and VQGAN-LC. We will make the VQ-VAE-based comparison more explicit in the revision.
> Second, we agree that hyperspherical latent-space methods are an important related direction. However, they are not a strict tokenizer baseline, since they generally learn continuous latent representations rather than discrete codebook/token representations for downstream token modeling. In the revision, we will clarify this distinction and better position ArcVQ-VAE relative to this line of work.
>
> >W2. Effect of forcing M(t) = 1 or changing the schedule shape
>
> Thank you for this important comment. To directly assess the role of the norm-bound schedule, we compare the original ArcVQ-VAE with a variant that keeps ArcLoss unchanged but fixes M(t)=1 for all iterations.
>
> **MNIST**
>
> | Method | l1 loss (↓) | PSNR (↑) | SSIM (↑) | LPIPS (↓) | rFID (↓) |
> |---|---:|---:|---:|---:|---:|
> | VQ-VAE | 0.0207 | 26.48 | 0.9777 | 0.0282 | 3.43 |
> | ArcVQ-VAE(M(t) = 1) | 0.0197 | 27.47 | 0.9786 | 0.0257 | 1.97 |
> | ArcVQ-VAE | 0.0175 | 28.13 | 0.9843 | 0.0213 | 1.59 |
>
> **CIFAR10**
>
> | Method | l1 loss (↓) | PSNR (↑) | SSIM (↑) | LPIPS (↓) | rFID (↓) |
> |---|---:|---:|---:|---:|---:|
> | VQ-VAE | 0.0527 | 23.32 | 0.8595 | 0.2504 | 39.67 |
> | ArcVQ-VAE(M(t) = 1) | 0.0489 | 24.01 | 0.8797 | 0.2112 | 30.70 |
> | ArcVQ-VAE | 0.0449 | 24.71 | 0.8976 | 0.1928 | 27.17 |
>
> As shown in the table above, fixing M(t)=1 throughout training still improves performance over vanilla VQ-VAE, indicating that the norm constraint itself is beneficial. However, the original time-dependent schedule, $ M(t)=\exp(\alpha t) $, achieves better performance than the fixed-M(t) variant. This suggests that while a constant bound stabilizes training, it becomes overly restrictive in later stages. In contrast, the time-dependent schedule provides stronger regularization early on while allowing greater representational flexibility later, leading to better overall performance.
>
> >Q1. What is the impact on LDM vs other posterior resampling models?
>
> Thank you for this important question. Our method is not tied to a specific downstream sampler. In the current submission, we used LDM as a representative large-scale latent generative model to evaluate tokenizer quality, but ArcVQ-VAE itself modifies the tokenizer rather than the sampler.
> To test transfer across sampler, we additionally conducted a VQ-VAE-based experiment on the CIFAR-10 dataset using an autoregressive PixelCNN sampler, comparing a vanilla VQ-VAE tokenizer and an ArcVQ-VAE tokenizer under the same downstream setup.
>
> | Method | Token Size | Codebook Size | FID (↓) |
> |---|---|---|---|
> | PixelCNN(VQ-VAE) | 64 | 512 | 57.36 |
> | PixelCNN(ArcVQ-VAE) | 64 | 512 | 43.13 |
>
> As shown in the table, replacing the vanilla tokenizer with ArcVQ-VAE again improves the downstream result under the same PixelCNN architecture and training protocol. This suggests that the benefit of ArcVQ-VAE is not specific to LDM alone. Rather, a stronger tokenizer can provide better discrete latent representations that are beneficial across different classes of downstream samplers operating on tokenized latents.
> We will include this comparison explicitly in the revision and clarify that our claim is tokenizer-side generality across multiple posteriors, not dependence on one particular sampler.
>
> >Q2. Can you use the “dictionary learning”-like representation in the latent space to avoid training a posterior sampler? If yes, how?
>
> Thank you for this insightful question. We would like to keep the claim careful. In the current paper, ArcVQ-VAE still follows the standard VQ-VAE pipeline and therefore still relies on a downstream sampler over discrete tokens. We do not claim that our method removes the need for posterior sampling.
> Our additional PixelCNN result supports a narrower conclusion: ArcVQ-VAE remains beneficial when the downstream sampler is changed. While improved tokenization may make latent modeling easier, our current evidence does not establish that ArcVQ-VAE eliminates the need for a downstream sampler. We will clarify this scope in the revision and present sampler removal as an interesting future direction rather than a claim of the present work.

---

> > ### Author Rebuttal · Reviewer_89GT · 2026-04-03
> >
> > I would have appreciate to have comparisons with methods which use the “dictionary learning”-like representation in the latent space to see the impact of choosing the VQ-VAE base to this work.
> > This is not completely convincing me of the importance and potential impact of the method.

---

> > > ### Author Response · Authors · 2026-04-05
> > >
> > > We thank the reviewer for this valuable suggestion. To better understand the impact of choosing the VQ-VAE framework relative to the general continuous latent-space case, we additionally compared four models under matched reconstruction settings: VAE, hyperspherical VAE, standard VQ-VAE, and ArcVQ-VAE.
> > >
> > > For fairness, the hyperspherical VAE was implemented with the same encoder/decoder architecture as VQ-VAE, differing only in the latent-space construction: instead of vector quantization with a discrete codebook, it uses a von Mises-Fisher posterior and a hyperspherical uniform prior following the standard hyperspherical VAE formulation.
> > >
> > > As shown in the Table below, the hyperspherical VAE consistently improves over the vanilla VAE, confirming that hyperspherical latent regularization is itself beneficial. However, it remains substantially worse than standard VQ-VAE on both MNIST and CIFAR-10, while ArcVQ-VAE achieves the best overall performance across all reported reconstruction metrics.
> > >
> > > **MNIST**
> > >
> > > | Method | l1 loss (↓) | PSNR (↑) | SSIM (↑) | LPIPS (↓) | rFID (↓) |
> > > |---|---:|---:|---:|---:|---:|
> > > | VAE | 0.0503 | 18.20 | 0.8505 | 0.0918 | 15.34 |
> > > | hyperspherical VAE | 0.0487 | 18.71 | 0.8720 | 0.0913 | 12.87 |
> > > | VQ-VAE | 0.0207 | 26.48 | 0.9777 | 0.0282 | 3.43 |
> > > | **ArcVQ-VAE** | **0.0175** | **28.13** | **0.9843** | **0.0213** | **1.59** |
> > >
> > > **CIFAR10**
> > >
> > > | Method | l1 loss (↓) | PSNR (↑) | SSIM (↑) | LPIPS (↓) | rFID (↓) |
> > > |---|---:|---:|---:|---:|---:|
> > > | VAE | 0.1135 | 17.45 | 0.5609 | 0.5120 | 97.76 |
> > > | hyperspherical VAE | 0.1057 | 17.48 | 0.6830 | 0.5459 | 89.51 |
> > > | VQ-VAE | 0.0527 | 23.32 | 0.8595 | 0.2504 | 39.67 |
> > > | **ArcVQ-VAE** | **0.0449** | **24.71** | **0.8976** | **0.1928** | **27.17** |
> > >
> > > This comparison is intended to isolate the framework-level effect of choosing the VQ-VAE formulation. Since the hyperspherical non-VQ baseline does not naturally provide a discrete tokenizer for downstream token-based generation, we focus this comparison on reconstruction quality and representation structure.
> > >
> > > More specifically, this comparison shows that a continuous latent-space method with a hyperspherical, dictionary-like latent structure is still weaker than a discrete VQ tokenizer, and that the strongest performance is obtained when hyperspherical geometry is imposed within a discrete codebook-based tokenizer. In other words, standard VQ-VAE already provides a substantial advantage over the general continuous latent-space case, and ArcVQ-VAE further improves this discrete representation by regularizing its codebook geometry on the hypersphere.
> > >
> > > We will include this comparison in the revision and clarify more explicitly that this is an important reason why the VQ-based formulation is beneficial in our setting.

---

### Official Review · Reviewer_zRrV · 2026-03-10

**Soundness:** 2
**Presentation:** 2
**Significance:** 2
**Originality:** 2
**Overall Recommendation:** 4
**Confidence:** 2

**Summary:**

This paper introduces ArcVQ-VAE, a variant of the Vector Quantized Variational Autoencoder (VQ-VAE) that performs quantization on a hypersphere rather than in Euclidean space. The authors argue that standard Euclidean distance leads to codebook collapse and poor utilization. By normalizing latent vectors and codebook entries to a unit sphere ($\|\mathbf{z}\| = 1, \|\mathbf{e}\| = 1$) and using arc-length (geodesic) distance for the commitment and codebook losses, the proposed method aims to improve representation stability and codebook efficiency. The method is evaluated on MNIST and CIFAR-10 datasets.

**Compliance With Llm Reviewing Policy:**

Affirmed.

**Ethical Review Concerns:**

Thanks for the detailed rebuttal and experiments. My concerns have been adequately addressed, and i will raise my score to 4.

**Final Justification:**

Thanks for the detailed rebuttal and experiments. My concerns have been adequately addressed, and i will raise my score to 4.

**Key Questions For Authors:**

1. Ablation vs. Cosine Similarity: Since arc-length is $\arccos(\text{cosine similarity})$, have you compared ArcVQ-VAE against a model that simply uses cosine similarity for quantization? What is the specific benefit of the $\arccos$ warp?
2. Scalability: How does ArcVQ-VAE perform when the latent dimension $D$ or the codebook size $K$ is significantly increased (e.g., $D=512, K=8192$)? Does the spherical constraint lead to training instability in high dimensions?
3. Comparison to EMA: Many practitioners use EMA updates to maintain codebook health. How does ArcVQ-VAE compare to a standard VQ-VAE equipped with EMA and codebook dead-word resetting?

**Limitations:**

yes

**Strengths And Weaknesses:**

Strengths:
1. Geometric Intuition: Transitioning from Euclidean to spherical geometry is a motivated approach to address the "dead code" problem by ensuring all vectors reside on a compact manifold.
2. Codebook Utilization: The empirical results show an improvement in codebook usage compared to the vanilla VQ-VAE baseline.

Weaknesses:
1. Limited Technical Novelty: The concept of normalized or spherical VQ is not entirely new. Mathematically, arc-length distance $d_{\text{arc}}(\mathbf{z}, \mathbf{e}) = \arccos(\mathbf{z}^\top \mathbf{e})$ is a monotonic transformation of cosine similarity. The paper fails to sufficiently distinguish why the arc-length formulation provides a fundamental advantage over existing cosine-similarity-based quantization or simple $\ell_2$ normalization techniques already common in the literature.
2. Insufficient Benchmarking: The evaluation is restricted to MNIST and CIFAR-10. For a top-tier conference like ICML, validation on small-scale datasets is insufficient. The community expects results on larger-scale benchmarks (e.g., ImageNet, FFHQ) or high-resolution generative tasks to prove the scalability of the spherical constraint.
3. Missing Competitive Baselines: The paper primarily compares against the "vanilla" VQ-VAE. It ignores modern, highly competitive variants designed to solve codebook collapse, such as Factorized Multi-Head VQ, Residual VQ (RVQ), or Exponential Moving Average (EMA) updates with restarts.
4. Without these comparisons, the claim of "superiority" is not well-supported.Theoretical Gap: While the paper claims improved stability, it lacks a rigorous gradient analysis showing how the arc-length loss behaves differently from Euclidean loss during the straight-through estimation (STE) backpropagation.

---

> ### Author Rebuttal · Authors · 2026-03-31
>
> Thank you for your constructive feedback. We appreciate your comments and respond to your concerns below.
>
> >W1 & Q1. Novelty relative to cosine similarity / benefit of the arccos warp?
>
> We agree that l2-normalized or spherical quantization alone is not new, and we do not claim novelty from that alone. Eq. (9) already shows that, after l2-normalizing latent vectors and codebook vectors in Eq. (8), codebook selection is equivalent to nearest-neighbor search under cosine similarity. Our contribution is therefore the combination of Ball-Bounded Norm Regularization in Eqs. (4) to (7), the additive angular-margin objective in Eqs. (10) to (13), especially Eq. (11), and the stop-gradient formulation in Eq. (12).
>
> The angular margin makes normalized cosine-based matching stricter: positives use $ \cos(\theta_{i,j} + m) $, while negatives use $\cos(\theta_{i,j})$. Since cosine is monotonically decreasing on $[0, \pi]$, $m>0$ lowers positive logits unless tighter angular alignment is achieved, encouraging stronger alignment to assigned codebook entries and separation from others. Table 5 already shows that BBNR improves over vanilla VQ-VAE and ArcLoss gives further gains.
>
> To verify that the gain does not come from cosine similarity alone, We additionally compared vanilla VQ-VAE, cosine-similarity matching, and ArcVQ-VAE under the same setting.
>
> **CIFAR10**
>
> | Method | PSNR (↑) | SSIM (↑) | LPIPS (↓) | rFID (↓) |
> |---|---:|---:|---:|---:|
> | VQ-VAE | 23.32 | 0.8595 | 0.2504 | 39.67 |
> | VQ-VAE(cosine-sim) | 23.37 | 0.8607 | 0.2466 | 39.04 |
> | ArcVQ-VAE | 24.71 | 0.8976 | 0.1928 | 27.17 |
>
> These results show that cosine matching alone yields only marginal changes, whereas ArcVQ-VAE gives clear gains across all metrics.
>
> >W2 & Q2. Evaluation beyond MNIST/CIFAR-10 / scalability
>
> We respectfully believe there is a misunderstanding here. The submission is not limited to MNIST and CIFAR-10: Sec. 5.1 states that ImageNet-1K is the primary dataset; Table 2 reports ImageNet-1K reconstruction at S = 16^2 and S = 32^2; Table 3 reports class-conditional ImageNet-1K generation with an LDM prior; Figure 7 and Appendix Figure 8 provide qualitative ImageNet results; and Table 4 reports a codebook-dimension ablation on FFHQ. We agree that these large-scale results were not emphasized clearly enough and will revise Sec. 5.1 and Sec. 5.2 accordingly.
>
> Regarding scalability, Table 4 already shows stable behavior across D = 4, 8, 16 on FFHQ, and all ImageNet experiments use 256^2 inputs, 32^2 latent tokens, D = 16, and K = 1024. We agree, however, that the current paper does not include a dedicated very-large-K scaling study, so we will keep that claim narrower.
>
> >W3 & Q3. Missing competitive baselines / comparison to modern tokenizer variants and EMA
>
> We agree that competitive baselines should be made more visible. The current submission already includes several non-vanilla baselines: Table 2 compares against ViT-VQGAN, RQ-VAE, MoVQ, SeQ-GAN, DF-VQGAN, CVQ-VAE, and VQGAN-LC; and Table 3 compares against VQGAN-FC (LDM), VQGAN-EMA (LDM), VQGAN-LC (LDM), RQVAE (RQ-Transformer), MoVQ (MaskGIT), and CVQ-VAE (LDM).
>
> Additionally, we compared ArcVQ-VAE against the suggested methods under the same token size setting.
>
> | Method | Token Size | Codebook Size | Usage | rFID (↓) |
> |---|---:|---:|---:|---:|
> | VQGAN | 256 | 1024 | 44% | 7.94 |
> | RQ-VAE | 256 | 16384 | - | 3.20 |
> | VQGAN-FC | 256 | 16384 | 11.2% | 4.29 |
> | VQGAN-EMA | 256 | 16384 | 83.2% | 3.41 |
> | **ArcVQ-VAE** | **256** | **1024** | **99%** | **1.63** |
>
> As shown above, even with a substantially smaller codebook, ArcVQ-VAE remains favorable in this setting. We will include this comparison explicitly.
>
> >W4. Theoretical gap / lack of rigorous gradient analysis under STE
>
> We agree that the current submission does not provide a full gradient-level comparison between the arc-length-based objective and Euclidean VQ loss under STE, and we do not claim theorem-level superiority. Our intended scope is empirical and mechanistic.
>
> That said, the current paper already includes two optimization-specific points that we will make more explicit. First, Eq. (12) applies stop-gradient to the codebook vectors inside ArcLoss, so this term backpropagates only through encoder outputs, while the codebook is still updated through the standard VQ objective in Eq. (3). Second, Appendix A gives a probabilistic interpretation of Eq. (18): ArcLoss can be viewed as a codebook-centric binary softmax objective, where top-k latent tokens are positives and the rest are negatives. The additive angular margin then encourages tighter positive alignment and stronger separation from negatives.
>
> We agree, however, that this remains optimization intuition rather than a full STE gradient analysis. We will therefore revise the wording so that claims about stability, representation quality, and geometry remain empirical. Figures 2, 3, and 5 and Table 5 provide empirical support consistent with this interpretation.

---

> > ### Author Rebuttal · Reviewer_zRrV · 2026-04-05
> >
> > Thanks for the detailed rebuttal and experiments. My concerns have been adequately addressed, and i will raise my score to 4.

---

> > > ### Author Response · Authors · 2026-04-05
> > >
> > > Thank you for the thoughtful follow-up and for raising the score. We greatly appreciate it and will reflect the feedback in the revision.

---

### Official Review · Reviewer_af1k · 2026-03-11

**Soundness:** 3
**Presentation:** 3
**Significance:** 3
**Originality:** 3
**Overall Recommendation:** 4
**Confidence:** 4

**Summary:**

This paper proposes ArcVQ-VAE, a VQ-VAE framework with a spherical angular-margin prior on the codebook. The method combines a norm regularization term and an angular margin loss to encourage better codebook dispersion and utilization. Experiments on MNIST, CIFAR-10, and ImageNet show improved reconstruction quality and stronger codebook usage, and the method also gives competitive results in latent diffusion based generation.

**Compliance With Llm Reviewing Policy:**

Affirmed.

**Key Questions For Authors:**

See weaknesses.

**Limitations:**

See weaknesses.

**Strengths And Weaknesses:**

Strengths
1. The paper addresses an important and practical problem with vector quantization: codebook under-utilization and poor geometric structure. The proposed idea is simple, intuitive, and easy to integrate into existing VQ-VAE style frameworks.
2. The empirical results are solid overall. The paper shows consistent improvements in reconstruction metrics and codebook usage across multiple datasets, and the generation results on ImageNet are also competitive.
3. The ablation study is reasonably informative and shows that both the norm regularization and the angular margin contribute to the final performance.

Weaknesses
1. The empirical gains over CVQ-VAE also appear somewhat marginal in some settings. While ArcVQ-VAE is generally competitive, the improvement over CVQ-VAE is not consistently strong across all datasets and metrics. In some cases, ArcVQ-VAE is even slightly worse. This is particularly noticeable on the ImageNet-1K reconstruction task, where under the same SS and KK setting, the reported rFID does not show a clear advantage over CVQ-VAE. This makes it harder to conclude that the proposed spherical angular-margin design yields a consistently stronger tokenizer than prior geometry-aware quantization methods.
2. Some of the paper’s interpretations are slightly stronger than the evidence supports. In particular, the claim that the method leads to semantically richer codebooks is mostly supported by indirect evidence such as usage statistics and visualization, rather than more direct semantic evaluation.
3. Some design choices are still heuristic. For example, the top-k selection in ArcLoss and the scheduling strategy for the loss weights seem reasonable, but the paper does not provide much justification for why these choices are preferred over alternatives.

---

> ### Author Rebuttal · Authors · 2026-03-31
>
> We sincerely thank the reviewer for the thoughtful and constructive feedback. We appreciate the positive overall assessment and agree that our claims, especially relative to CVQ-VAE and codebook quality, should be stated more carefully. We respond to your concerns as follows.
>
> >W1. The gains over CVQ-VAE are sometimes marginal
>
> We agree that the draft should not imply uniform superiority over CVQ-VAE. The results are mixed: in Table 1, ArcVQ-VAE improves over CVQ-VAE on MNIST across all metrics, while on CIFAR-10 it improves l1, PSNR, SSIM, and LPIPS but is worse on rFID. In Table 2 under the same ImageNet-1K reconstruction setting with S=256 and K=1024, ArcVQ-VAE reaches 99% usage with rFID 1.63, while CVQ-VAE reports 100% usage with rFID 1.57.
>
> Our claim is therefore narrower: ArcVQ-VAE is competitive overall, with especially favorable results at higher token resolution and in downstream generation. Table 2 shows rFID 0.95 at S=1024 with K=1024, and Table 3 shows FID 6.79 in LDM-based ImageNet generation versus 6.87 for CVQ-VAE at the same codebook size. We will revise the paper accordingly and remove wording that suggests universal superiority.
>
> We also clarify that our claim is not that ArcVQ-VAE wins by maximizing usage alone. The evidence instead suggests that tokenizer quality depends not only on usage but also on redundancy among active entries. This is consistent with Table 2, where CVQ-VAE reaches 100% usage and VQGAN-LC reaches 99% usage with a 100000-entry codebook, while ArcVQ-VAE still attains the best reported reconstruction result at S=1024 with K=1024. We will revise the paper to present ArcVQ-VAE more precisely as a competitive tokenizer design achieving a favorable tradeoff between codebook usage and downstream quality under a small codebook budget.
> As additional supporting evidence, we conducted a post-hoc reduction experiment on VQGAN-LC. Starting from a trained VQGAN-LC model with K = 100000, we reduced the codebook size by k-means clustering and re-evaluated reconstruction.
>
> Table: (R) denotes the reduced-codebook variant obtained via k-means clustering from the pretrained original codebook.
> | Method | Codebook Size | Usage | rFID (↓) |
> |---|---:|---:|---:|
> | VQGAN-LC | 100000 | 99.5% | 1.13 |
> | VQGAN-LC (R) | 50000 | 99.5% | 1.13 |
> | VQGAN-LC (R) | 16384 | 99.7% | 1.13 |
> | VQGAN-LC (R) | 8192 | 99.9% | 1.14 |
> | VQGAN-LC (R) | 4096 | 100% | 1.26 |
> | VQGAN-LC (R) | 1024 | 100% | 2.40 |
> | VQGAN-LC (R) | 32 | 100% | 79.2 |
>
> As shown above, reconstruction remains nearly unchanged down to K = 8192 while usage stays near 100%. Although this does not replace a direct method comparison, it supports the point that large codebook size and high usage alone do not fully determine tokenizer quality, since many active entries can still be redundant.
>
> >W2. Some interpretations, especially “semantically richer codebooks,” are stronger than the evidence supports
>
> We agree with this point. The submission does not provide a direct semantic benchmark. Instead, the evidence is indirect but meaningful: Figure 3 shows a brighter and more homogeneous pairwise l2-distance structure, indicating less redundancy; Figure 5 shows more spatially structured quantized latent maps; and Tables 1–3 show improved reconstruction and competitive generation. We will therefore replace stronger semantic wording with claims directly supported by the evidence, such as “more discriminative and less redundant codebook geometry,” “more spatially structured quantized representations,” or “stronger reconstruction/generation performance under a small codebook budget.” The revised wording will remain empirical and observational.
>
> >W3. The top-k design and the scheduling of the loss weights are heuristic
>
> We agree that these choices are heuristic. For top-k in Eq. (11), however, Table 6 already shows that varying $k \in {1, 3, 8}$ causes only small changes on MNIST and CIFAR-10, indicating that the method is not sensitive to the exact choice of k. Eq. (11) also makes the role of $N_j^{(k)}$ clear: it defines a local positive set for each codebook entry so that ArcLoss aligns nearby latents while treating the rest as negatives. Thus, top-k is a practical local-neighborhood construction; the heuristic part is the default choice k = 3. We will clarify this in Sec. 4.3 and Sec. 5.1.
> For the loss-weight schedule, we agree that the justification is more limited. Eq. (13) uses a larger $\gamma(t)$ early in training and decays it later so reconstruction can dominate. This is a practical training heuristic rather than a theoretically optimal schedule, and the current submission does not compare alternative schedules. We will make this scope explicit and clarify in Sec. 5.1 and Appendix B that $s = 10, m = 0.1, k = 3, \gamma_0 = 1.0$, and the reported $\lambda$ values are fixed pragmatic settings, while Table 6 specifically supports robustness for k, m, and s. Alternative scheduling strategies will be left as future work.

---

> > ### Author Rebuttal · Reviewer_af1k · 2026-04-03
> >
> > I thank the authors for their thorough response. After reading the rebuttal, I consider my previous concerns to be fully addressed. In my view, the paper offers useful insight, and the reported performance improvements further support its value. The overall contribution remains clear and technically meaningful. Based on the quality of the paper and the improvements over other baselines, I will keep my original rating unchanged.

---

> > > ### Author Response · Authors · 2026-04-05
> > >
> > > We sincerely thank the reviewer for the thoughtful comments. We truly appreciate that our rebuttal has addressed the reviewer’s concerns. The feedback was highly valuable in improving the paper, and we will faithfully reflect these points in the revision.

---

### Official Review · Reviewer_cEzp · 2026-03-11

**Soundness:** 2
**Presentation:** 3
**Significance:** 2
**Originality:** 2
**Overall Recommendation:** 5
**Confidence:** 4

**Summary:**

The paper adds regularisation terms to spread out latent representations learned under a VQ-VAE encoder/decoder framework, as used in generative models such as VQ-VAE, VQ-GAN and latent diffusion models. Empirical results appear to support the heuristic design choices.

**Compliance With Llm Reviewing Policy:**

Affirmed.

**Final Justification:**

I believe the content of the paper merits publication and the suggested improvements following the review address presentational concerns.

I increased my score 4 --> 5

**Key Questions For Authors:**

* The justification of the paper is rather hand-wavy and does not always come across as being well justified. The paper would read more robustly if this were improved, e.g.
    - 062: "geometric *imbalance* of the latent space" presupposes there should be "balance" - how defined/justified?
    - 099: "ensuring uniformity by providing additional margins on the hypersphere has been shown to be effective in representation learning," - needs citation/support
    - 144(R): "we observe a *distinct imbalance* in the .2-norms of the learned codebook vectors." - again, what "balance", no theoretical justification is provided, there seems no obvious way to know that imbalance isn't *useful* for performance.
    - 162(R): "*excessive* clustering", as previous - how is this objectively "excessive"?
    - 239: "thereby *more faithfully* reflecting the spherical structure of the latent space" - faithful in what sense, again there is not particular reason to know this is "better", this section simply describes using cosine similarity, a well-known but poorly understood heuristic. Also it is not clear that the latent space is "inherently spherical".
   Overall, the regularisation approach taken seems reasonable based on prior work and analysis/observations of latent representations learned by previous methods. The approach taken appears justified by its empirical results, which suggests that the changes induced to the representation distribution helped. But without theoretical backing, it seems unsound to make claims about properties of the representations a priori.
* There are no error bars/ranges on any of the results, how significant is the variance over different runs?

**Limitations:**

See strengths/weaknesses

**Strengths And Weaknesses:**

Strengths
 * The paper's main strength lies in the empirical results, which show improved generative performance relative to baselines.
* The paper is relatively easy to read and contains a good amount of ablation on hyperparameter choices and the loss components.

Weaknesses
* Model description - Section 3 does not fully describe the generative model in question, only how to encode/decode a single "patch" to/from latent space. That is not the end in itself and it should be mentioned that
    (a) a *vector* of encodings is learned (dimensionality depending on data type, e.g. 2-D grid for images), hence the representation of a data sample, once discretised, is a vector of codebook indices; and
    (b) the distribution over those discrete vectors must be learned, e.g. by pixel-CNN (in VQ-VAE), transformer, diffusion process, etc.
  - this is relevant, not only because that is the model and motivation, but since by training the encoder to create a more complex distribution of representations, presumably makes it a harder job to learn that distribution (here diffusion).

* The definition of the loss terms could be simpler/more succinct, e.g. Eq 6, simply clips vector magnitudes, Eq 7 is obvious/redundant given that, Eq 8, are simply "unit vectors".
* Eq 10/11 - it would be clearer to state that $\theta$ is always positive, otherwise cosine is maximal when $\theta+m=0$, i.e. $\theta=-m$.
* Since the optimal theta is 0, for any $m$ (including 0), it is unclear why $m>0$ helps (I did not find the appendix clarified much).
* LDM is never defined.

---

> ### Author Rebuttal · Authors · 2026-03-31
>
> We thank the reviewer for the thoughtful and constructive feedback. We appreciate the positive assessment of the empirical results and ablations, and agree that some claims should be stated more carefully.
>
> >W1 & W2 & W5.
>
> We sincerely thank the reviewer for these helpful comments and suggestions. We agree that these issues concern presentation rather than the core method, but they are important for clarity. In the revision, we will clarify the full generative pipeline in Section 3 by explicitly describing image -> spatial token grid -> downstream prior over discrete tokens. We will also streamline Eqs. (6)–(8) by presenting Eq. (6) as the norm-bounding update, Eq. (7) as its geometric interpretation, and Eq. (8) as the normalization used in the angular formulation. We also agree that LDM should be defined explicitly, and we will define it as Latent Diffusion Model at first mention.
>
> >W3 & W4. Clarify the angle domain in Eq. 10/11 and why the additive margin helps
>
> We agree that Eq. (10) should explicitly state that $\theta_{i,j} = \arccos(\hat{z}_i(x)^\top \hat{e}j)$ lies in $[0, \pi]$, removing the ambiguity in Eqs. (10) and (11). We also agree that the role of m in Eq. (11) should be explained more rigorously in the main text. The purpose of m > 0 is not to change the optimum, which still occurs at $\theta = 0$, but to impose a stricter criterion on positive pairs. Since the positive logits use $\cos(\theta{i,j} + m)$, the negative logits use $\cos(\theta{i,j})$, and cosine is monotonically decreasing on $[0, \pi], m > 0$ lowers the positive logits unless tighter angular alignment is achieved. We will revise the main text around Eq. (11) to make this reasoning explicit.
>
> >Q1.
>
> We agree that some statements in the current draft are stronger than the evidence directly supports. Our intended scope is empirical rather than theorem-level, and we will revise the wording accordingly.
>
> >062 & 144(R) “geometric imbalance” / “imbalance in the l2-norms”
>
> We agree that “balance” is under-defined. We do not claim that there is a theoretically optimal notion of perfectly balanced codebook norms. Rather, we use “imbalance” operationally to describe the norm skew shown in Fig. 2 and Sec. 4.1: frequently selected codebook vectors accumulate larger norms, while rarely selected or unused vectors remain near initialization with much smaller l2 norms. Our claim is therefore not that all codebook vectors should have identical norms, but that this extreme norm disparity is a practically observable failure mode associated with under-utilization and reduced effective coverage. We will revise the wording accordingly.
>
> >099 citation/support for the hypersphere-uniformity statement
>
> We agree that the current sentence is broader than what the cited works directly establish. Our purpose was to motivate Eq. (11), not to claim a general theoretical result about all margin-based hyperspherical methods. We will narrow this sentence, make its connection to Eq. (11) explicit, and add more targeted support.
>
> >162(R) “excessive clustering” / 239 “more faithfully reflecting the spherical structure of the latent space”
>
> We agree that both phrasings are too strong. Our point is not that there is a universal threshold for “excessive” clustering, but that the learned codebook shows an empirically observable concentration-and-redundancy pattern that reduces effective coverage; Figure 3 provides a concrete basis for this point. Likewise, after l2-normalizing both latent vectors and codebook vectors as in Eq. (8), the matching rule in Eq. (9) is naturally interpreted in terms of angular similarity on the unit hypersphere. This is a restricted geometric interpretation of the normalized quantization rule, not a claim that the latent space is intrinsically spherical or theoretically optimal. We will revise the wording accordingly.
>
> >Q2. Variability across runs
>
> We agree that reporting variability across runs would strengthen the empirical section. We therefore evaluated a representative setting across multiple runs and found that the variance is small relative to the observed performance gap, so the main conclusion remains unchanged.
>
> **MNIST**
>
> | Run | Method | l1 loss (↓) | PSNR (↑) | SSIM (↑) | LPIPS (↓) | rFID (↓) |
> |---|---|---|---|---|---|---|
> | 1 | ArcVQ-VAE | 0.0175 | 28.13 | 0.9843 | 0.0213 | 1.59 |
> | 2 | ArcVQ-VAE | 0.0176 | 28.12 | 0.9843 | 0.0214 | 1.61 |
> | 3 | ArcVQ-VAE | 0.0176 | 28.10 | 0.9842 | 0.0215 | 1.66 |
> |  | Mean ± Std | 0.0175 ± 0.0001 | 28.12 ± 0.02 | 0.9843 ± 0.0001 | 0.0214 ± 0.0001 | 1.62 ± 0.04 |
>
> **CIFAR10**
>
> | Run | Method | l1 loss (↓) | PSNR (↑) | SSIM (↑) | LPIPS (↓) | rFID (↓) |
> |---|---|---|---|---|---|---|
> | 1 | ArcVQ-VAE | 0.0449 | 24.71 | 0.8976 | 0.1928 | 27.17 |
> | 2 | ArcVQ-VAE | 0.0452 | 24.67 | 0.8965 | 0.1935 | 28.12 |
> | 3 | ArcVQ-VAE | 0.0450 | 24.69 | 0.8970 | 0.1934 | 27.77 |
> |  | Mean ± Std | 0.0450 ± 0.0002 | 24.69 ± 0.02 | 0.8970 ± 0.0006 | 0.1932 ± 0.0004 | 27.69 ± 0.48 |

---

> > ### Author Rebuttal · Reviewer_cEzp · 2026-04-03
> >
> > I have scanned the other reviews and see no significant issues.  While the paper combines notions mostly seen elsewhere (as do many published works), it does so logically in a way that broadly improves performance.
> >
> > I believe the paper merits publication based on its content and the suggested improvements following the review will improve its presentation.
> >
> > I increase my score 4 --> 5

---

> > > ### Author Response · Authors · 2026-04-05
> > >
> > > We sincerely thank the reviewer for the insightful comments and thoughtful suggestions. We greatly appreciate the updated assessment. These remarks were very helpful in clarifying the scope of our claims and strengthening the overall paper. We will carefully reflect these points in the revision.

---

### Decision · Program_Chairs · 2026-04-30

**Decision:**

Accept (regular)

**Comment:**

Based on the four reviews and the authors’ rebuttal, I recommend **Accept**. The paper introduces a practical and well-motivated improvement to VQ-VAE via spherical angular-margin prior and norm regularization, leading to better codebook utilization and reconstruction quality. All technical concerns—including limited novelty, marginal gains over CVQ-VAE, missing baselines, and evaluation scope—were thoroughly addressed in the rebuttal with additional experiments (e.g., comparison to cosine similarity, EMA, VQ-VAE, hyperspherical VAE, and PixelCNN downstream tasks).